# Grain refinement in titanium prevents low temperature oxygen embrittlement

Yan Chong ®[1,2,3,4], Reza Gholizadeh ®[2], Tomohito Tsuru ®[3,5] ✉,
Ruopeng Zhang ®[1,4], Koji Inoue[6], Wenqiang Gao[7], Andy Godfrey ®[7],
Masatoshi Mitsuhara[8], J. W. Morris Jr. ®[1], Andrew M. Minor ®[1,4] ✉ &
Nobuhiro Tsuji ®[2,3] ✉

Interstitial oxygen embrittles titanium, particularly at cryogenic temperatures, which necessitates a stringent control of oxygen content in fabricating titanium and its alloys. Here, we propose a structural strategy, via grain refinement, to alleviate this problem. Compared to a coarse-grained counterpart that is extremely brittle at 77 K, the uniform elongation of an ultrafine-grained (UFG) microstructure (grain size ~ 2.0 μm) in Ti-0.3wt.%O is successfully increased by an order of magnitude, maintaining an ultrahigh yield strength inherent to the UFG microstructure. This unique strength-ductility synergy in UFG Ti-0.3wt.%O is achieved via the combined effects of diluted grain boundary segregation of oxygen that helps to improve the grain boundary cohesive energy and enhanced <c + a> dislocation activities that contribute to the excellent strain hardening ability. The present strategy will not only boost the potential applications of high strength Ti-O alloys at low temperatures, but can also be applied to other alloy systems, where interstitial solution hardening results into an undesirable loss of ductility.

Interstitial oxygen has long been considered as a double-edged sword in pure titanium with a hexagonal close packed (hcp) structure. Oxygen can significantly increase the strength[1–8], but it can also severely deteriorate the ductility[4–8], rendering titanium extremely brittle at 77 K when the oxygen content merely exceeds 0.3 wt.%[7,8]. The dramatic decrease in ductility with increasing oxygen content is also accompanied with a transition from transgranular ductile fracture to intergranular brittle fracture[7]. Consequently, a full harnessing of the potent strengthening effect of oxygen interstitials is limited, and instead, it is necessary to maintain tight control of the oxygen content during the manufacturing of titanium which in turn leads to its high cost[9–11]. Moreover, the catastrophic failure of Ti-O alloys at 77 K greatly hinders the potential applications of this high strength to weight material as structural

material in liquid-propellant rocket engines, or as liquid nitrogen/helium vessel materials, since in both cases good low-temperature ductility and toughness have been put as the priority requirement due to the safety concern. The mechanistic basis for this detrimental oxygen sensitivity of ductility in titanium has been largely attributed to a localized plastic deformation (in the form of planar slip and {11$\bar{2}$4} deformation twinning) and vulnerable grain boundaries[7], both of which are inherent deformation characteristics of high oxygen containing titanium.

Here, we propose a fundamental strategy to overcome this dilemma in hcp-titanium via grain refinement. It is demonstrated in ultrafine-grained (UFG) (average grain sizes $D$ ~ 2.0 μm, close to the upper limit of what can be considered as UFG structure) titanium containing 0.3 wt.% oxygen that an ultrahigh strength (~1250 MPa) and

[1]Department of Materials Science and Engineering, University of California, Berkeley, CA, USA. [2]Department of Materials Science and Engineering, Kyoto University, Kyoto, Japan. [3]Elements Strategy Initiative for Structural Materials (ESISM), Kyoto University, Kyoto, Japan. [4]National Center for Electron Microscopy, Molecular Foundry, Lawrence Berkeley National Laboratory, Berkeley, CA, USA. [5]Nuclear Science and Engineering Center, Japan Atomic Energy Agency, Tokai-mura, Ibaraki, Japan. [6]Institute for Materials Research, Tohoku University, Oarai, Ibaraki, Japan. [7]Key Laboratory of Advanced Materials (MoE), School of Materials Science and Engineering, Tsinghua University, Beijing, China. [8]Department of Advanced Materials Science and Engineering, Kyushu University, Fukuoka, Japan. ✉e-mail: tsuru.tomohito@jaea.go.jp; aminor@berkeley.edu; tsuji.nobuhiro.5a@kyoto-u.ac.jp

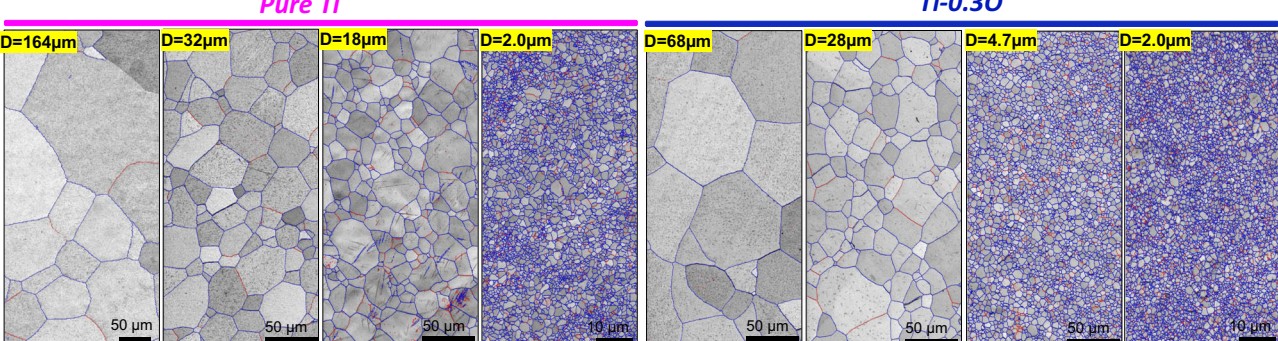

**Fig. 1 | Initial microstructures of pure Ti and Ti-0.3O alloys.** Representative grain boundary maps (blue lines: high-angle grain boundaries with misorientation angle, $\theta > 15°$, red lines: low-angle grain boundaries with misorientation angle, $2° < \theta \le 15°$) of pure Ti and Ti-0.3O with different average grain sizes ($D$) obtained by high-pressure torsion (HPT) and annealing.

large ductility (~14%) can be simultaneously achieved at 77 K (Fig. 1), avoiding the catastrophic brittle fracture that typically occurs in the coarse-grained (CG) counterparts ($D \ge 50.0\,\mu m$). The underlying mechanism for the elimination of oxygen embrittlement in UFG titanium is three-fold. First, the propensity of intergranular cracking is substantially restricted by diluting the grain boundary segregation of oxygen (via a substantial increase in total grain boundary area), which thereby increases the grain boundary cohesive energy in the UFG microstructure. Second, the localized plastic deformation inside the grains is greatly mitigated by activating $<c+a>$ dislocations, owing to the high stress level inherent to deformation of an UFG microstructure. Lastly, the fracture is retarded due to an equal proportion of intergranular and transgranular micro-cracks/voids in the UFG microstructure. It is also worth noting that this overall strategy can be applied to other systems where interstitial solution hardening results into an undesirable loss of ductility.

## Results and discussion
### Tensile properties
In order to refine the grain size, severe plastic deformation (high-pressure torsion) is applied to the initial coarse-grained microstructure followed by annealing. As shown by the grain boundary maps in Fig. 1 (blue lines: high-angle grain boundaries (HAGBs) with misorientation angle, $\theta > 15°$; red lines: low-angle grain boundaries (LAGBs) with misorientation angle, $2° < \theta \le 15°$), fully recrystallized microstructures with different average grain sizes ($D$: 2.0–164 μm) were successfully obtained in both pure Ti (as a reference) and the Ti-0.3O alloy. In addition, the Ti-0.3O alloy generally has a smaller grain size than pure Ti for the same annealing condition (Supplementary Fig. 1), which can be attributed to the drag effect of oxygen interstitials on the migration of grain boundaries[12–14].

The engineering stress-strain curves of pure Ti and Ti-0.3O alloy at both room temperature, 300 K, (red lines) and liquid nitrogen temperature, 77 K, (blue lines) are shown in Fig. 2a, b, respectively. Both alloys exhibited a higher yield strength at a smaller grain size and lower deformation temperature. However, the grain size dependence of uniform elongation for the two alloys at 77 K was opposite, as shown by the statistical plots (six specimens per grain size for each alloy) in Fig. 2c, d. (Note the different strain regimes for the two alloys.) In pure Ti, even though the UFG microstructure still exhibited excellent strain-hardening at 77 K (Fig. 2a), the uniform elongation gradually decreased (from 63% to 40%) with decreasing grain size (Fig. 2c), following the strength-ductility trade-off in metallic materials[15–17]. In the Ti-0.3O alloy, the CG microstructure had a very limited elongation ($\varepsilon_f \sim 1.5\%$) at 77 K (Fig. 2b), similar to the value we previously reported[7,8]. Surprisingly however, the uniform elongation greatly increased from 1.5% to 14.0% with decreasing grain size (Fig. 2d), together with a simultaneous

increase in tensile strength from 1.0 GPa to 1.25 GPa (Fig. 2b). Such a remarkable increase in tensile ductility of UFG Ti-0.3O at 77 K not only boosts the potential applications of high strength Ti-O alloys at low temperatures, but also sheds light on the underlying mechanism for the abrupt ductile to brittle transition in Ti-O alloys with increasing oxygen content at 77 K. In line with the increased tensile ductility, fracture tomography of the Ti-0.3O alloy also showed a clear transition from intergranular brittle fracture in the CG microstructure to transgranular dimpled rupture in the UFG microstructure (Fig. 2e and Supplementary Fig. 2). This suggests a fundamental change of deformation behavior in Ti-0.3O caused by grain refinement. In comparison, the fracture tomography of pure Ti exhibited dimple patterns regardless of the grain size (Fig. 2e).

### Mesoscopic deformation behaviors
To explore the mechanistic basis of the anomalously increased tensile ductility in the UFG Ti-0.3O alloy at 77 K, we start from a comparison of the mesoscopic deformation behavior in pure Ti and Ti-0.3O with both coarse and ultrafine grain sizes (Figs. 3 and 4). For the CG microstructures, there was a huge gap in tensile ductility between the two alloys (Fig. 3a). Compared with the continuous strain-hardening and large uniform elongation ($\varepsilon_u \sim 60\%$) of CG ($D = 120\,\mu m$) pure Ti that mainly originated from the large number of deformation twins (Fig. 3b), CG ($D = 68\,\mu m$) Ti-0.3O fractured shortly after yielding ($\varepsilon_f \sim 1.5\%$). From a detailed characterization of the microstructure near the fracture surface (Fig. 3c, e), it is revealed that the early fracture of the CG Ti-0.3O specimen was due to easy formation of grain boundary cracks (indicated by yellow arrows in Fig. 3c) and their quick intergranular propagation. Interestingly, most of these grain boundary cracks were connected with deformation twins, implying an adverse effect of deformation twins in this alloy system. Based on the EBSD method developed by Wilkinson[18–20], the local residual strain fields of two representative regions (*region A* and *B* marked in Fig. 3c) were calculated from the high-resolution EBSD data, as shown in Fig. 3d. Comprehensive results including all six strain tensors of the two regions are given in Supplementary Fig. 3b, c. In *region A*, where the propagation of a {11$\bar{2}$4} twin was blocked at the grain boundary, a significant residual strain field was revealed near the grain boundary area, which could be one of the reasons for preferential crack initiation at twin boundary (TB) / grain boundary (GB) intersections. In *region B*, several planar slip bands were observed (Fig. 3c, e), a typical dislocation slip mode in Ti-0.3O at low temperature[7,8]. The planar slip bands also led to localized deformation, but associated with much lower strain magnitude and an alternating distribution characteristic (Fig. 3d). It should also be noted that the high stress at TB/GB intersections could also trigger the formation of nano-twins near the GB (e.g., the rectangular area in Fig. 3e), which helps to relax the stress

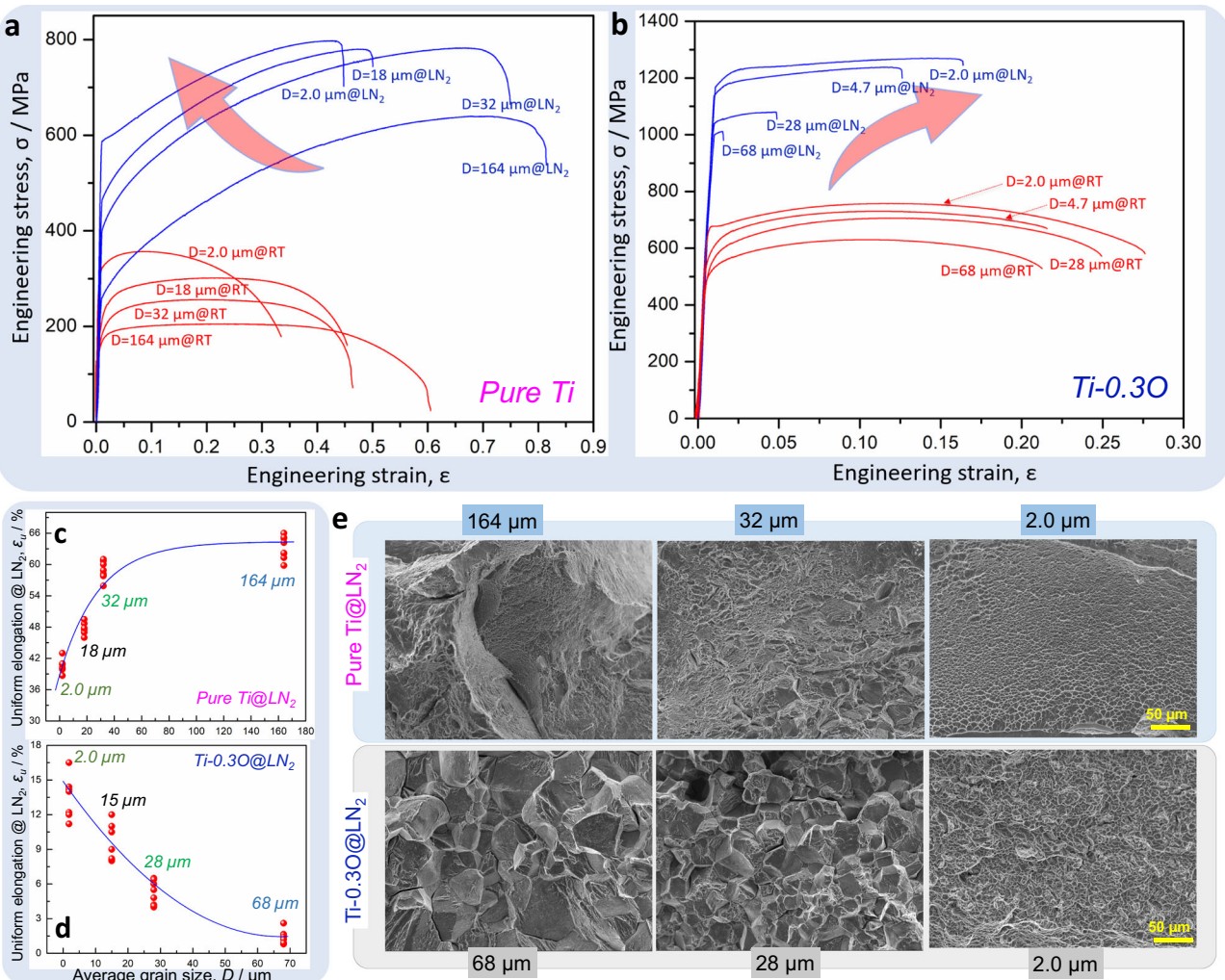

**Fig. 2 | Mechanical properties of pure Ti and Ti-0.3O alloys with different grain sizes.** Engineering stress-strain curves of pure Ti **a** and Ti-0.3O **b** with different grain sizes at room temperature (red curves) and liquid nitrogen temperature (blue curves). Note the different strain regimes (on the x-axis) for the two alloys. The evolution of uniform elongations at liquid nitrogen temperature with grain size in pure Ti **c** and Ti-0.3O **d**. Opposite evolution tendencies were found between the two alloys. **e** Typical fracture tomography of pure Ti and Ti-0.3O with different grain sizes at liquid nitrogen temperature.

localization and prevent GB cracks (see Supplementary Fig. 3d for details).

For samples with a UFG microstructure, the gap in tensile ductility between the two alloys at 77 K was significantly narrowed (Fig. 4a). As a matter of fact, the strain-hardening rate of UFG Ti-0.3O was even slightly higher than that of UFG pure Ti (insert in Fig. 4a), and its relatively lower ductility was an intrinsic outcome of a doubled yield strength. In UFG pure Ti, there were still a lot of deformation twins (Fig. 4b, c), although the fractions were lower (Fig. 4b) than in the CG counterparts. In UFG Ti-0.3O, however, deformation twins were totally suppressed (Fig. 4d), leaving dislocation activity as the sole contributor to the excellent strain-hardening ability of the material. Moreover, long grain boundary cracks were generally eliminated in the UFG Ti-0.3O alloy, and the plastic deformation became relatively more homogenous, as reflected by the EBSD kernel average misorientation (KAM) maps in Fig. 4d. Therefore, from a mesoscopic point of view, the improved tensile ductility in UFG Ti-0.3O can be attributed to both *reduced grain boundary cracking* and *homogeneous intra-grain plastic deformation*.

**Grain boundary chemistry**

The *reduced grain boundary cracking* in the UFG Ti-0.3O alloy can be interpreted from the perspective of grain boundary chemistry. To this end, two atom probe tomography (APT) specimens were prepared from grain boundary areas of the CG ($D = 68\,\mu m$) (Fig. 5a) and UFG ($D = 2.0\,\mu m$) Ti-0.3O samples (Fig. 5b) by a site-specific lift-out method using a focused-ion beam (FIB) instrument (see Materials and Methods for details). Transmission electron microscopy (TEM) images from the two specimens (Fig. 5c, d) before APT experiments explicitly confirmed the existence of grain boundaries near the APT specimen tips. The distances from the grain boundary planes to the tips were measured to be 115 nm and 210 nm for CG and UFG specimens, respectively. This distance enabled the acquisition of sufficient data to capture the grain boundaries via APT. In addition, the misorientation angles of the two grain boundaries were determined to be 57° and 70° (Fig. 5c, d), respectively. Therefore, APT results obtained from the two specimens can be considered representative of HAGBs in the two microstructures.

The oxygen atom maps obtained from the CG and UFG Ti-0.3O specimens are shown in Fig. 5e and f, respectively. Detailed results including the distributions of all trace elements are shown in Supplementary Fig. 4. In the CG Ti-0.3O specimen, there was a clear segregation of oxygen atoms around an inclined plane, located at a distance from the tip that perfectly matches that of the grain boundary in Fig. 5c. A similar result was also obtained in another CG Ti-0.3O APT specimen containing a larger portion of grain boundary that was

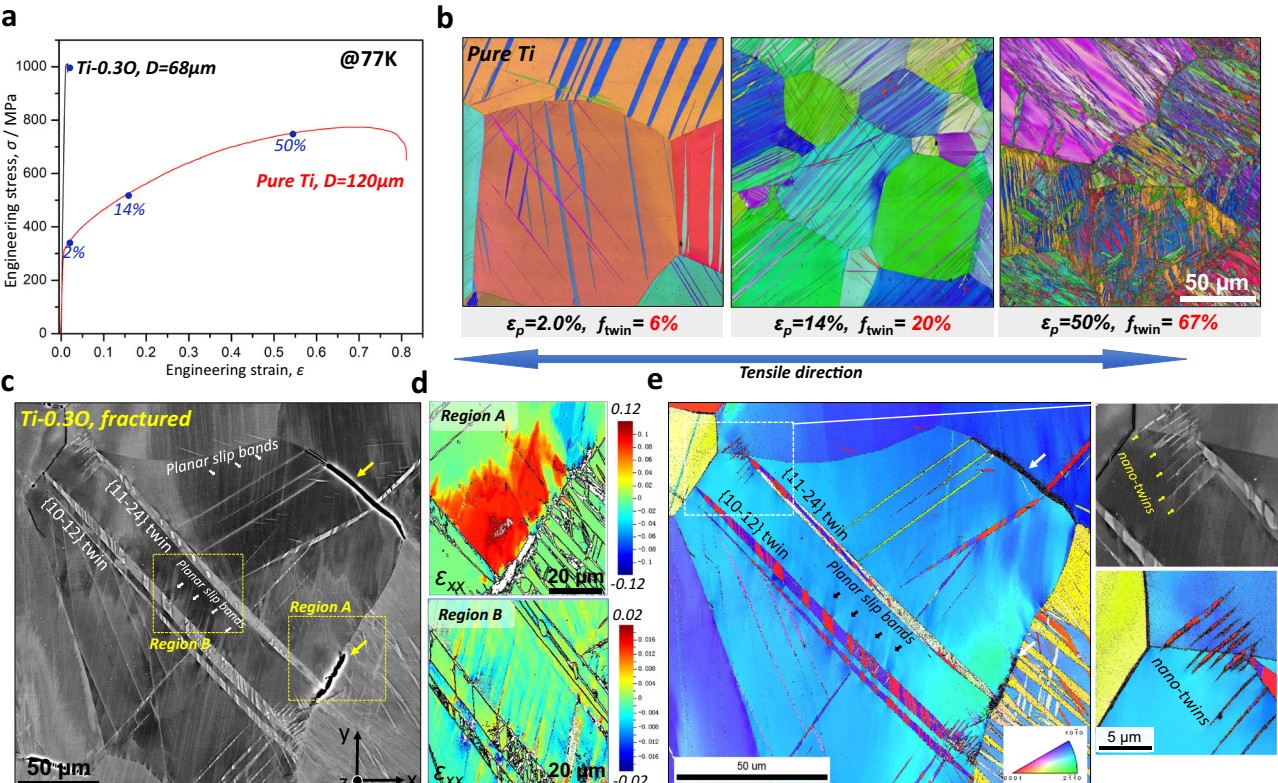

**Fig. 3 | Mesoscopic deformation behavior of CG pure Ti and Ti-0.3O at 77 K.**
**a** Engineering stress-strain curves of CG pure Ti and Ti-0.3O at 77 K. **b** EBSD inverse pole figure (IPF) maps of CG pure Ti at interrupted plastic strains of 2.0%, 14% and 50%. The fraction of deformation twins is given for each strain. **c** Back-scattered electron (BSE) image of CG Ti-0.3O after tensile fracture (1.5%), in which GB cracks and slip bands are indicated by yellow and white arrows, respectively. **d** Cross-correlation EBSD-derived residual strain field of two representative regions (indicated by rectangle in **c**) for TB/GB intersections and planar slip bands in CG Ti-0.3O. **e** IPF map of **c**, in which the activation of nano-twins at the TB/GB interaction is highlighted.

aligned almost parallel to the tip (see Supplementary Fig. 5). Thus, we can safely establish at this point that, there is indeed segregation of oxygen atoms near the HAGBs in the CG Ti-0.3O specimen. The elemental profiles (Fig. 5g) across the grain boundary (see Supplementary Fig. 4a and Methods for details) clearly reveal that the peak oxygen concentration near the grain boundary in the CG sample can reach 2.0 at.%, almost twice of that inside the grains (~1.1 at.%). It should be emphasized that the oxygen concentration inside the grains agrees well with the bulk oxygen concentration (1.0 at.%) of the specimen, which strongly confirms the accuracy of our APT results. To the best of our knowledge, this is the first report to date on grain boundary segregation of oxygen in HCP alpha-titanium, in line with a recent report of oxygen segregation in BCC beta-titanium alloy[21]. In addition, there is a strong segregation of other trace elements such as iron, a weak segregation of carbon and no clear segregation of nitrogen near the grain boundary of CG specimen (Supplementary Fig. 4b).

In contrast, no grain boundary segregation of oxygen was observed in the UFG Ti-0.3O specimen, as evidenced by the atom map (Fig. 5f) and oxygen concentration profile across the grain boundary (Fig. 5g). Carbon and nitrogen were also homogeneously distributed across the grain boundary (Fig. 5g). This phenomenon can be rationalized from a dilution effect if we consider the grain boundary plane as a potential segregation site for oxygen atoms. A reduction of grain size from 68 μm to 2.0 μm corresponds to increasing the grain boundary area per volume by 34 times, which would certainly diminish the average segregation level at each grain boundary and promote a homogenous distribution of oxygen atoms in the UFG Ti-0.3O specimen.

The origin of grain boundary segregation of oxygen in titanium was explored by first-principles calculations (see Methods for details).

The results of the interaction between solutes are summarized in Fig. 5h. At first, the stability of the interaction between solute elements (Al, V, Mo, Fe) and oxygen atoms was investigated as a fundamental feature of oxygen in Ti. Oxygen atoms have a repulsive interaction with all these major alloying elements, as well as with vacancies in bulk regions (inside grains) (Interaction panel of Fig. 5h). The result indicates that oxygen atoms are not trapped around the major alloying elements in the bulk regions of Ti. Subsequently, the GB segregation of oxygen atoms was investigated. The $(10\bar{1}4)$ grain boundary with a misorientation angle of 50.5° about the $\langle 11\bar{2}0\rangle$ rotation axis, which are energetically unstable compared to the other twin and grain boundaries from preliminary calculations, was taken as suitable examples of HAGB. The segregation energy for interstitial oxygen and substitutional solutes were evaluated, where all possible sites for both interstitial and substitutional sites were considered, with the average values shown in the Segregation panel of Fig. 5h. It is shown that oxygen does not segregate spontaneously at HAGB (indicated by a large positive segregation energy, 0.6 eV), while Fe solutes exhibit a strong tendency for grain boundary segregation (manifested by a large negative segregation energy, −0.61 eV). The effect of segregation on the cleavage behavior was subsequently calculated, where $\Delta 2\gamma_{int}$ is defined as the difference in the cleavage energies between Ti-O and pure Ti. The value of $\Delta 2\gamma_{int}$ is an excellent indicator to estimate the effect of oxygen on the ideal fracture toughness of the interface[22]. According to this indicator, oxygen atom behave as a strong embrittlement factor at the interface since the cleavage energy significantly decreases. The other trace elements, e.g. nitrogen and carbon, despite of a weak segregation tendency at the grain boundary, were believed to play a minor role in affecting grain boundary coherency of Ti-0.3O alloy, due to their considerably lower bulk concentrations in the material. There remains

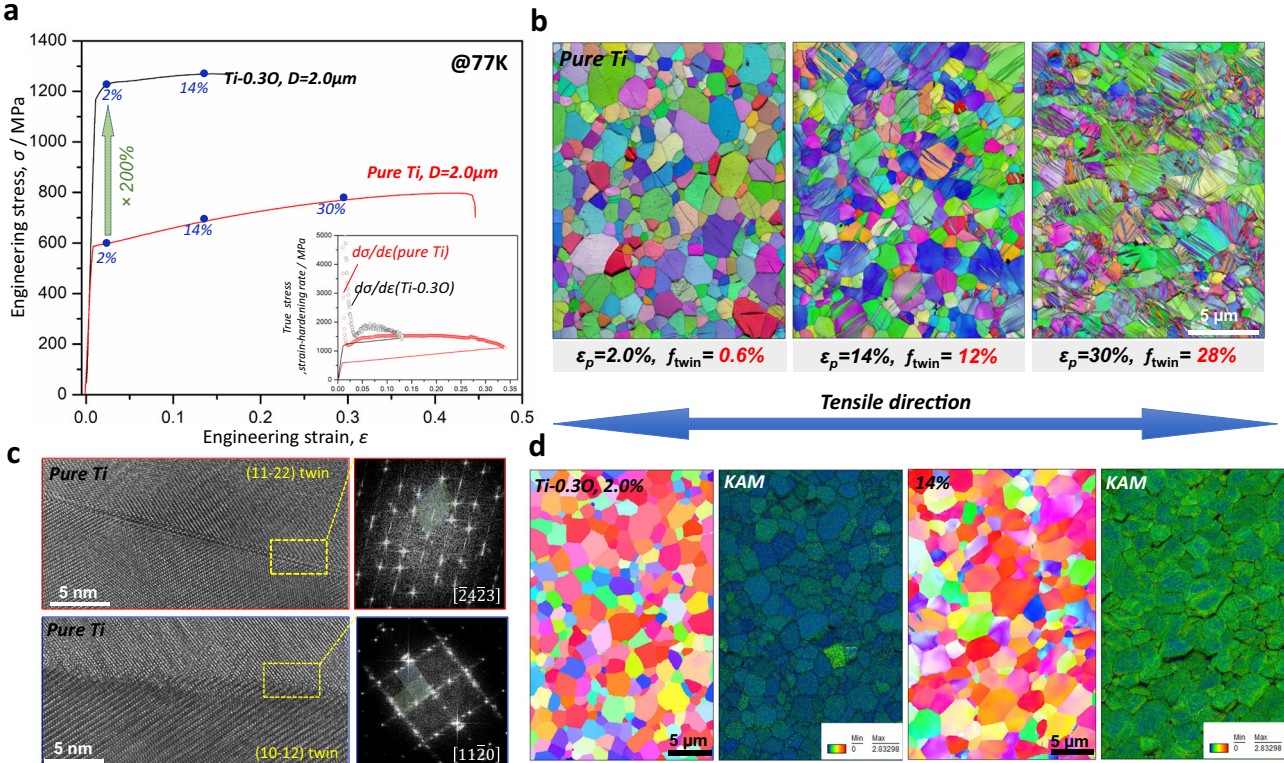

**Fig. 4 | Mesoscopic deformation behavior of UFG pure Ti and Ti-0.3O at 77 K.** **a** Engineering stress-strain curves of UFG pure Ti and Ti-0.3O at 77 K. The true stress-strain and strain-hardening rate curves of the two alloys are inserted. **b** IPF maps of UFG pure Ti at interrupted plastic strains of 2.0%, 14% and 30%. The fractions of deformation twins are much smaller than those in CG pure Ti. **c** High resolution TEM images together with fast Fourier transformation (FFT) patterns of $\{11\bar{2}2\}$ and $\{10\bar{1}2\}$ twins in UFG pure Ti at an interrupted strain of 12%. **d** IPF maps of UFG Ti-0.3O at interrupted plastic strains of 2.0% and 14%, together with the EBSD kernel average misorientation (KAM) maps. No deformation twins were observed in UFG Ti-0.3O after tensile deformation.

an essential question of why oxygen atoms were observed around grain boundary regions in many experiments. As shown in the segregation panel of Fig. 5h, a high positive segregation energy (-0.6 eV) was revealed for oxygen atoms residing at the grain boundary, indicating that oxygen atoms segregating at the grain boundary *alone* is not energetically favorable in Ti. However, the effect of major substitutional elements at the grain boundary is the key to clarify this problem. Segregation of oxygen around the iron solute at the grain boundary was examined, where interestingly, it was revealed that oxygen segregates stably, due to the attractive interaction with iron only at grain boundary regions, even though oxygen atoms are not, by themselves, stable at a grain boundary. Presumably, there are more stable configurations of Fe-O pairs at grain boundaries than in the bulk. Furthermore, it should be noted that the change in the cleavage energy when considering Fe-O pair segregation at the grain boundary became lower than in the case of considering O segregation alone at the grain boundary, although in both cases the segregation tended to decrease the cleavage energy, i.e. deteriorate the grain boundary cohesivity.

The theoretical calculations on grain boundary cohesive energies are also supported by experimental evidence, in which the grain boundary cohesive/bonding of CG microstructures with different oxygen contents are qualitatively compared. Tensile deformations of CG pure Ti, Ti-0.1O and Ti-0.3O alloys at 77 K were all interrupted at the fracture stress of Ti-0.3O (-1000 MPa), corresponding to a true strain of 38% for pure Ti and 26% for Ti-0.1O (Supplementary Fig. 6a). It is clearly shown that, despite the abundant deformation twins formed at the high stress level, the grain boundaries remained intact in both pure Ti (Supplementary Fig. 6d) and Ti-0.1O (Supplementary Fig. 6c) without any appreciable grain boundary cracks/voids. This is in sharp contrast to the high propensity of grain boundary crack formation in

the CG Ti-0.3O alloy (indicated by yellow arrows in Supplementary Fig. 6b). Such a direct comparison of grain boundaries at the same stress level clearly demonstrates that a higher tendency of grain boundary segregation will greatly decrease the grain boundary cohesive energy, rendering the grain boundaries in high oxygen specimens as intrinsically brittle.

## Enhanced <c+a> dislocation activity

Given that the grain boundaries become strong enough to withstand the high yield stress in UFG Ti-0.3O without early grain boundary cracking, the alloy has the ability to sustain extensive dislocation plasticity inside the grains, providing sufficient strain-hardening rate (Fig. 4a) to sustain the high stress level after yielding and allowing a large uniform elongation (-14%) to be achieved. A detailed TEM characterization of dislocation activity in the CG and UFG Ti-0.3O specimens at 77 K was also carried out (Fig. 6). In CG Ti-0.3O, well-developed planar slip behavior along prismatic planes was observed after tensile fracture at 77 K (Fig. 6a), consistent with our previous results[7]. The dislocation pile-ups can be as long as several tens of micrometers and impose a high stress level at grain boundaries. In addition, this planar dislocation slip represents localized plastic deformation (as revealed by the local strain field in Fig. 3d) and substantially reduced possibility of cross-slip. In UFG Ti-0.3O, on the other hand, dislocations were homogenously distributed inside the ultrafine grains and planar dislocations slip was not observed regardless of the plastic strain (Fig. 6b and Supplementary Fig. 7). This phenomenon indicates another contribution of grain refinement to the homogenous plastic deformation in UFG Ti-0.3O alloy. That is a total suppression of the deleterious planar slip simply by the decrease of grain size, under the same deformation conditions where otherwise long planar slip bands would

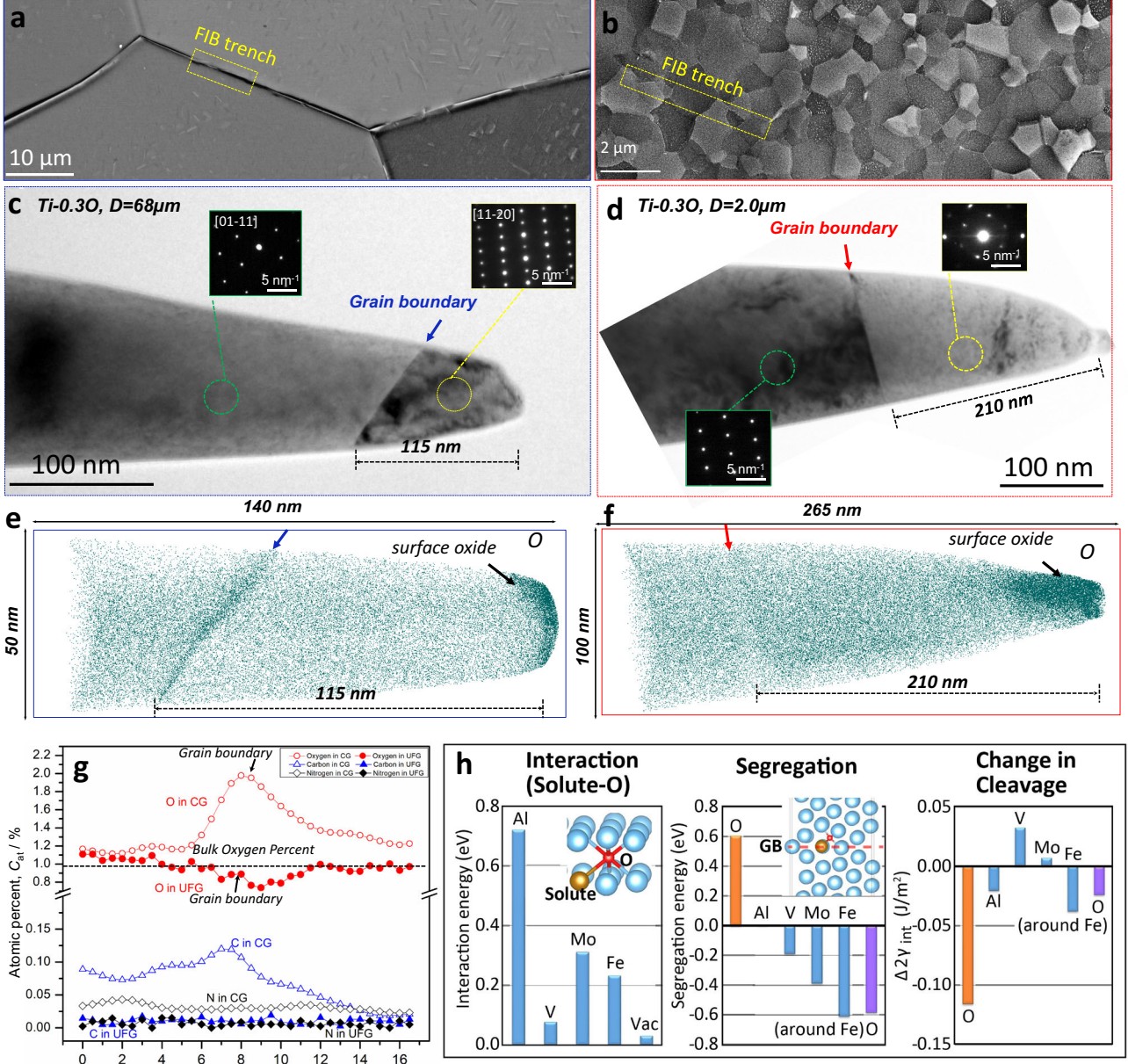

**Fig. 5 | GB chemistry analysis of CG and UFG Ti-0.3 O.** BSE images of CG ($D = 68\,\mu m$) **a** and UFG ($D = 2.0\,\mu m$) **b** Ti-0.3O. Schematic illustrations of the grain boundary lift-out positions for APT specimens are indicated in both microstructures. Transmission electron microscopy (TEM) images of APT specimens prepared from CG **c** and UFG **d** Ti-0.3O samples. Grain boundaries near the tips are clearly visible and confirmed by diffraction patterns. (In the APT specimen from CG sample **c**, grain orientations of [01$\bar{1}$1] and [11$\bar{2}$0] were confirmed, whereas in the APT specimen from UFG sample **d**, grain orientations of [1$\bar{2}$13] and close to [0001] were determined.) Oxygen atom (green color) maps of CG **e** and UFG **f** samples, in which the locations of grain boundaries are indicated by arrows according to the TEM images. Clear oxygen segregation near the grain boundary was observed in the CG Ti-0.3O sample, in contrast to the more homogenous distribution of oxygen in the UFG Ti-0.3O sample. **g** Compositional profiles of interstitial elements (O, C and N) across grain boundaries in the CG and UFG Ti-0.3O samples. **h** First-principles calculation results of interaction between oxygen and solutes, oxygen segregation energy along a HAGB, as well as its effect on grain boundary cleavage energy ($\Delta 2\gamma_{int}$).

be predominant in the coarse-grained microstructure. One of the reasons for the suppressed planar slip in UFG Ti-0.3O could be the limited grain interior space, far less enough for the development of planar slips, as well as a larger fraction of grain boundaries which generally disturb the propagation of planar slip. A detailed dislocation tomography analysis (Fig. 6a-i and b-i) of deformed CG and UFG Ti-0.3O specimens (see Supplementary Movies 1–4) also confirmed the localized dislocation pattern in the CG specimen and the more homogenously distributed dislocations in the UFG specimen in 3-dimensional space. This is in a good agreement with the uniform distribution of KAM values in UFG Ti-0.3O deformed to a similar plastic strain (Fig. 4d). Thus, both mesoscopic and microscopic microstructural characterizations undoubtedly demonstrate a more homogeneous plastic deformation in the UFG Ti-0.3O alloy.

Furthermore, <*c* + *a*> dislocations were confirmed to be present inside the ultrafine grains of Ti-0.3O at a plastic strain of 2.5% (Fig. 6c), according to a "*g* dot *b*" analysis under two-beam conditions. Unlike the occasionally observed <*c* + *a*> dislocations only near grain boundaries in the CG Ti-0.3O specimens,[7] the density of <*c* + *a*> dislocations in the UFG Ti-0.3O specimens was much larger, and the dislocations tended to be distributed inside the grains (Fig. 6c-i and **iii**). In addition, the <*a*> dislocations in the UFG Ti-0.3O specimen

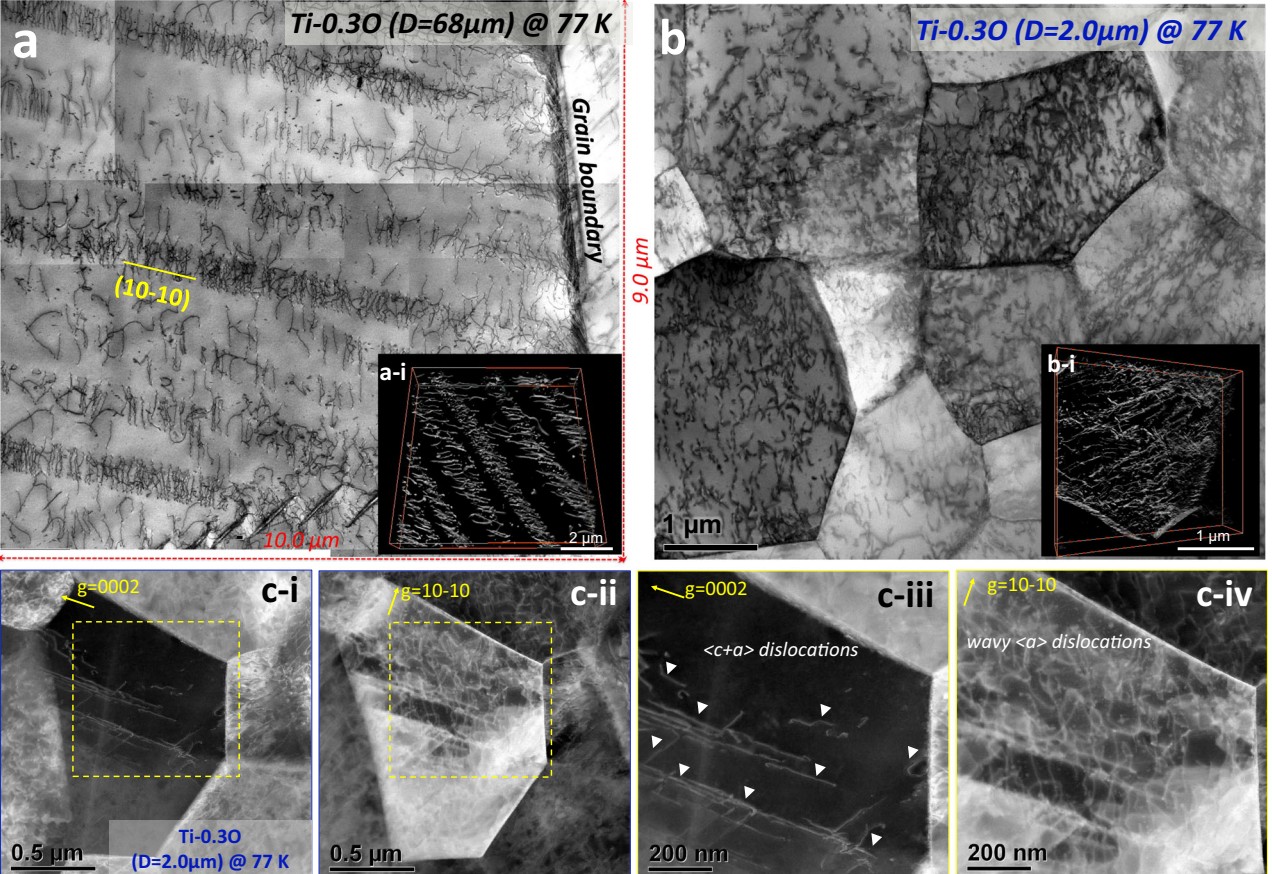

**Fig. 6 | TEM characterization of dislocations in CG and UFG Ti-0.3O deformed at 77 K. a** Well-aligned planar dislocation slip along (10-10) prismatic planes in the CG Ti-0.3O alloy after tensile fracture ($\varepsilon_f \sim 1.5\%$) at 77 K. Inserted is a snapshot of dislocation tomography analysis, which confirms a planar dislocation arrangement in 3-dimensional space. **b** In the UFG Ti-0.3O alloy deformed at 77 K by a plastic strain of 2.5%, dislocations were more homogeneously distributed inside the grains. In the inserted snapshot of dislocation tomography analysis, planar slip bands are also absent. Videos of the dislocation tomography data for both samples are included in the Supplementary Videos. **c** Two-beam condition analysis of dislocations in UFG Ti-0.3O deformed by 2.5% at 77 K. A large portion of <**c** + **a**> dislocations (**c-i** and **c-iii**) is confirmed inside the ultrafine grains, interacting with <**a**> dislocations in wavy configurations (**c-ii** and **c-iv**).

exhibited highly curved configurations (Fig. 6c-ii and iv), in sharp contrast to the straight and parallel aligned <**a**> dislocations in the CG Ti-0.3O specimen. The formation of abundant <**c** + **a**> dislocations can also be accounted for the absence of planar slips in the UFG Ti-0.3O specimen. As a comparison, the Burgers vector of dislocations in UFG pure Ti at the same plastic strain (Supplementary Fig. 8) was also examined, where it was found that <**a**> dislocations were predominant (Supplementary Fig. 8) and <**c** + **a**> dislocations were only scarcely detected (Supplementary Fig. 8a, c).

The high propensity of <**c** + **a**> dislocations, presumably due to the ultrahigh yield stress level, can contribute to a large uniform elongation of the UFG Ti-0.3O microstructure in three perspectives. Firstly, due to a total suppression of deformation twins, <**c** + **a**> dislocations serve as the only deformation mode that accommodates plastic strain along the <**c**> direction. Thus, a high density of <**c** + **a**> dislocations plays a crucial role in achieving a generalized plastic deformation and arbitrary shape change of the specimen. Secondly, these <**c** + **a**> dislocations, gliding on {11$\bar{2}$2} pyramidal planes, have a higher chance of interacting with <**a**> dislocations gliding on {10$\bar{1}$0} planes, leading to the accumulation of a number of tangled dislocations and resulting in large strain-hardening homogeneously, as shown in Fig. 6c-iv. Finally, the replacement of planar dislocation slip by homogenously distributed <**c** + **a**> and <**a**> dislocations can effectively reduce the local stress concentrations at grain boundaries caused by dislocation pile-ups. The overall effect is to delay the formation of grain boundary cracks/voids in the UFG Ti-0.3O specimen.

The last benefit of grain refinement to the enhanced tensile ductility of UFG Ti-0.3O can be attributed to a tortuous crack propagation path. Compared to an easy crack propagation along the straight and long grain boundaries in the CG specimen (Supplementary Figs. 6b and 9b), intergranular crack propagation can be stopped after propagating a short distance at GB triple junctions in the UFG specimen, where as we observed, most cracks in this case remained as small voids (Supplementary Fig. 9d). Moreover, the possibility of transgranular cracks/voids becomes close to that of intergranular cracks/voids in the UFG specimen (Supplementary Fig. 9c, d). There is a higher chance for these transgranular cracks to be arrested inside the grains, as shown in Supplementary Fig. 9e, f, which helps to delay the fracture of the material.

To summarize, we proposed a simple and cost-effective strategy to solve the long-standing oxygen embrittlement issue in titanium via grain refinement, without sacrificing the ductility even at 77 K. Fundamentally, the synergy of ultrahigh tensile strength and large ductility in ultrafine-grained microstructures is realized through the combined effects of reducing detrimental grain boundary segregation of oxygen, and promoting <**c** + **a**> dislocations inside the grains. This strategy could also be applied to other alloy systems, where interstitial solution hardening results into an undesirable loss of ductility. Moreover, in light of the harmful effect of oxygen segregation on the grain boundary cohesive energy and tensile ductility, a potential grain boundary chemistry engineering approach can be proposed for Ti-O alloys. By doping the alloy with other trace elements (e.g. boron and

yttrium) that have a stronger grain boundary segregation tendency than oxygen, the grain boundary cohesivity of Ti-O alloys could be either enhanced or further decreased, leading to an improved or even deteriorated tensile ductility of the material. This, however, requires further experimental and modeling work with respect to each specific doping element.

## Methods

### Materials processing and mechanical property testing

Two Ti ingots with different oxygen impurity contents (0.018 wt.% and 0.31 wt.%, corresponding to 0.057 at.% and 0.95 at.%, respectively) were received. They will be referred to as pure Ti and Ti-0.3O alloy hereafter for simplicity. The detailed chemical compositions are shown in Supplementary Table I. In addition, a coarse-grained ($D = 80\,\mu m$) Ti-0.1O alloy (oxygen content of 0.10 wt.%, corresponding to 0.31 at.%) was used as a reference. The reason for choosing Ti-0.3O as the targeting alloy was based on our previous study,[7] in which a critical oxygen content of 0.30 wt.% was established for causing a ductile-to-brittle transition in Ti-O alloys at 77 K. Both pure Ti and Ti-0.3O ingots were homogenized at 850 °C for 2 h followed by water quenching to room temperature. Discs for high-pressure torsion (HPT) (10 mm in diameter and 0.8 mm in thickness) were then prepared by the use of electrical discharge machining (EDM). HPT experiments were carried out under a pressure of 6.0 GPa with a rotation speed of 0.5 rotation per minute at room temperature. All discs were deformed by 5 rotations, and the equivalent strain $\varepsilon_{eq}$ at the edge of the disc was calculated to be ~200, based on the equation of $\varepsilon_{eq} = 2\pi r n/\sqrt{3}h$, where $r$ is the distance from the center, $n$ the number of rotations and $h$ the thickness of the specimen. After the HPT deformation, the specimens were annealed at 480-880 °C for 600 s followed by water quenching to room temperature. Additionally, some HPT deformed samples were annealed at 610 °C for 30-3600 s also followed by water quenching to room temperature. All the heat treatments were carried out under a high vacuum condition ($\leq 2 \times 10^{-3}$ Pa) to reduce the oxidation as much as possible. Miniature-sized tensile specimens (gauge length: 2.0 mm, cross section: $1.0 \times 0.6\,mm^2$) were prepared from the discs by EDM in such a way that the center of the gauge part coincided with the position at a radial distance of 3.0 mm from the center. Uniaxial tensile tests were performed on a Shimadzu AG-X system with an initial strain rate of $8.3 \times 10^{-4}\,s^{-1}$ at either room temperature (~300 K) or liquid nitrogen temperature (~77 K). The plastic strain of room temperature tensile deformation was precisely measured by a digital image correlation (DIC) method. The details can be found in our previous publications[23,24]. In contrast, the liquid nitrogen temperature tensile deformation was conducted in a special container that allowed full immersion of both the miniature-sized tensile specimens and tensile jigs in the liquid nitrogen during the tensile tests. All the set-ups were kept in liquid nitrogen for 5 min before testing for temperature homogenization. The tensile strain of liquid nitrogen temperature tensile deformation was calibrated using the room temperature strain data (obtained by the DIC method), assuming a roughly constant Young's modulus at both temperatures[7,8]. It should be noted that the Young's modulus actually slightly increased with decreasing temperature (in some cases, the variations could be around 1%). However, this variation is comparatively smaller than the changes of total elongation caused by different deformation temperatures, grain sizes and oxygen contents, as seen in this study. For each grain size, three specimens were tested to guarantee reproducibility.

Due to the limited thickness (~0.6 mm) of the tensile specimens, only a few grains were included in the thickness direction of coarse-grained microstructures of pure Ti ($D = 164\,\mu m$) and Ti-0.3O ($D = 68\,\mu m$) alloys, which may cause uncertainties in the obtained mechanical properties, as they can hardly be considered as polycrystalline specimens. To this end, we also prepared some larger-sized tensile specimens (gauge geometry: 10 mm ($l$) × 2 mm ($w$) × 1.15 mm

($t$)) for the CG microstructures (where severe plastic deformation is not necessary) of both alloys, and compared their tensile properties at 77 K with those from micro-sized tensile specimens. The comparison is shown in Supplementary Fig. 10. It is clearly shown that the engineering stress-strain curves of specimens with different thickness were quite similar for both pure Ti and Ti-0.3O alloys with a coarse grain size. This confirms the reliability of the mechanical properties we obtained for CG microstructures from micro-sized tensile specimens.

### Sample preparation and characterization method

Cross-section surfaces at a radial distance of 3.0 mm from the disc center were mechanically polished, followed by electro-polishing (in a solution of 10% perchloric acid and 90% methanol) and final chemical etching in Kroll's reagent. Details of the sample preparation can be found elsewhere[7,8]. The microstructure observations, including scanning electron microscopy (SEM), back-scattered electron (BSE) and electron backscattered diffraction (EBSD) characterizations were conducted on a field emission gun (SEM) JEOL 7800F. The average grain sizes were measured by a line intercept method. The kernel average misorientation (KAM) maps were calculated considering the first-nearest neighbors of each measured point. For scanning transmission electron microscopy (STEM) investigations of the deformed microstructures, tensile deformation at liquid nitrogen temperature was interrupted at the plastic strains of 2.5%, 8.0%, and 12.0%. The thickness of deformed specimens was reduced to ~100 μm by mechanical polishing, and then followed by twin-jet electro-polishing using a Struers TenuPol-5 at -40 °C at a voltage of ~30 V, in a solution of 6% perchloric acid, 34% n-butanol and 60% methanol. STEM observations were conducted on a JEOL 2100 TEM operated at 200 kV. To determine the Burgers vectors of dislocations in UFG pure Ti and Ti-0.3O alloys after tensile deformation ($\varepsilon_p = 2.5\%$), '$\boldsymbol{g} \cdot \boldsymbol{b}$' analysis was conducted under a two-beam condition from the $[11\bar{2}0]$ zone axis. Dark-field images were taken using a $\boldsymbol{g}$ vector of either 0002 or $10\bar{1}0$, such that the Burgers vector can be confirmed using the invisibility criterion of $\boldsymbol{g} \cdot \boldsymbol{b} = 0$. High-resolution TEM characterization of deformation twins in UFG pure Ti at an interrupted plastic strain of 12% was conducted on a FEI TitanX 60-300 operated at an accelerating voltage of 300 kV. It should be noted that the postmortem TEM analysis at room temperature will not introduce any additional deformation and thus should not be considered to alter the morphologies of dislocations/twinning left by cryogenic deformation. In addition, we can only expect minor effects from the relaxation of internal stress caused by temperature change from 77 K to room temperature.

### Residual strain field

For investigation of residual strain using cross-correlation EBSD, the data were collected using a Tescan field-emission gun SEM operated at 20 kV. EBSD maps were collected at a working distance of 16.5 mm, a microscope magnification of 2000 and a 70° stage tilt, at a step-size of 1 μm over areas of $40 \times 50\,\mu m^2$ (Region A), $40 \times 40\,\mu m^2$ (Region B) or $45 \times 45\,\mu m^2$ (Region C). The EBSD patterns were collected using a CMOS-based detector (Oxford Instruments, Symmetry) at full resolution of $1244 \times 1024$ pixels (no binning), with a total collection time per pattern of 160 ms to ensure high quality patterns. Analysis of the EBSD patterns was carried out using the CrossCourt software package (BLG Vantage). For strain analysis the reference pattern was chosen as the map pixel with the smallest local average misorientation to its neighbors (with a manual confirmation that this was a pixel located in the same grain but away from the feature of interest). The cross-correlation calculations were carried out based on 20 overlapping regions of interest of size $256 \times 256$ pixels in each EBSD pattern, using filtering conditions of high frequency cut-off, high frequency width, low frequency cut-off, and low frequency width, of 2, 0, 36, and 18, respectively. The strains reported in this work are expressed relative to the tensile sample coordinate system, where ($x, y, z$) are parallel to the

tensile direction, transverse direction and sample normal direction, respectively.

## Atom probe tomography experiments

Atom probe tomography (APT) specimens including grain boundaries of both coarse-grained ($D = 68\,\mu m$) and ultrafine-grained ($D = 2.0\,\mu m$) Ti-0.3 O alloy were prepared by a site-specific lift-out method using a FEI Quanta 3D 200i dual-beam focused-ion beam (FIB) instrument. Metallographic samples were carefully prepared by electro-polishing and chemical etching (with Kroll's reagent) to clearly reveal the grain boundaries. A wedge lift-out geometry was used to mount multiple samples onto a copper coupon. Final tip sharpening was carried out at a voltage of 30 kV and beam current of 49 pA, followed by a tip polishing at a voltage of 5 kV and beam current of 49 pA. The APT specimens were quickly transferred to TEM and characterized before data acquisition. The location of grain boundary was confirmed by examining the SAD patterns on both sides. The distance between the grain boundary plane and the specimen tip was also measured. APT experiments were carried out on a local electrode atom probe (LEAP 4000) instrument from CAMECA Instruments using a laser mode under ultra-high vacuum ($<4 \times 10^{-11}$ torr). Data was collected at a temperature of 50 K, with a laser pulse energy of 50 pJ. The frequency was 160 kHz and detection rate was 0.5%. The data reconstruction was performed using Integrated Visualization and Analysis Software (IVAS) version 3.8.2. Elemental histograms across the grain boundaries were obtained from a cylinder perpendicular to the grain boundary plane with a step size of 0.5 nm.

In the data analysis, both $^{16}O^{+1}$ and $^{64}TiO^{+1}$ peaks were used to identify the spatial distribution of oxygen atoms. Since $^{16}O^{+1}$ and $^{48}Ti^{+3}$ peaks are overlapped in the mass spectrum, quantitative analysis of oxygen was performed by decomposing the overlapped components based on the intensity of the non-overlapped isotope peaks of Ti such as $^{46}Ti^{+3}$, $^{47}Ti^{+3}$, $^{49}Ti^{+3}$, $^{50}Ti^{+3}$ and the natural isotope abundance. However, the formation of titanium oxides (accounting for the $^{64}TiO^{+1}$ peak) inside the APT specimen was excluded based on the following reasons. Firstly, a surface titanium oxide layer (thickness less than 10 nm) was clearly observed near the tips of both CG and UFG specimens (Supplementary Fig. 4b, d), which probably formed during the sample transfer between FIB fabrication, TEM observation and APT data collection. The atomic percent of oxygen in the titanium oxide layers was as large as 5.0 at.%. Easy oxidation of tip areas (grain interiors) excluded the possibility of preferential oxidation only near the grain boundaries. Secondly, the APT tip evaporation and data acquisition were carried out in an ultra-high vacuum ($<4 \times 10^{-11}$ torr) and low temperature condition (50 K), which in our opinion, would reduce the oxidation as much as possible. Finally, in the elemental profiles obtained from both CG and UFG Ti-0.3O specimens (see Fig. 3g), the oxygen concentrations of grain interiors were close to the bulk oxygen concentration (~1.0 at.%) of the material, and were much lower than that (~5.0 at.%) of titanium oxides formed near the tips. This serves as a strong evidence that the oxygen distribution behavior obtained from our APT experiments (except the surface oxide layer) was NOT related to sample oxidation. Nevertheless, one possible reason for the frequent detection of oxygen atoms in the form of $^{64}TiO^{+1}$ complex could be due to a high affinity between oxygen and titanium. There is a high chance of oxygen and titanium ions forming $^{64}TiO^{+1}$ complex, during the flight process to the detector after being evaporated from the specimen. This, however, will not affect the grain boundary oxygen segregation results we obtained in the CG Ti-0.3O alloy, considering the equal opportunity of all oxygen atoms in forming $^{64}TiO^{+1}$ complex after evaporation.

## Dislocation tomography

Dislocation tomography analysis of deformed CG and UFG Ti-0.3O specimens was conducted through tilt-series observation on a Titan

Cubed G3 microscope (Thermo Fisher Scientific) at an accelerating voltage of 300 kV, under the BF-STEM mode. The convergence semi-angle of the incident beam was set at 10 mrad. In the tilt-series acquisition, the specimen-tilt axis was set to be exactly parallel to the diffraction vector, $10\bar{1}1$ for the CG specimen or $0\bar{1}1\bar{1}$ for the UFG specimen, by using a high-angle triple-axis holder (Mel-Build HATA-8075). STEM images were recorded every 2° over the tilt range of −60° to +60°. A 3D view of the dislocations was reconstructed by the weighed back projection algorithm by the Thermo Fisher Scientific's Inspect 3D software.

## Computational methods

**Construction of atomic models.** A 180-atom supercell with dimensions a $3 \times 3 \times 3$ along $[11\bar{2}0]$, $[\bar{1}100]$, and $[0001]$ axes were prepared for perfect hcp crystal model, where the lattice constants were set to be $a_0 = 2.936$ and $c/a = 1.5818$. Then two types of grain boundary configurations were considered to investigate the grain boundary segregation and the effect of solute on the fracture process. $(10\bar{1}4)$ and $(30\bar{3}2)$ grain boundaries were chosen as the example of unstable grain boundary because these grain boundaries have relatively higher grain boundary energy compared with other twin and grain boundary structures (Supplementary Fig. 11). The total number of atoms are 202 and 208 for the $(10\bar{1}4)$ and $(30\bar{3}2)$ grain boundaries, respectively. The major substitutional solute elements (Al, V, Mo, and Fe) and interstitial solutes (O) were introduced at these sites to calculate the segregation energy. The initial position of the interstitial site at the grain boundary region were predicted using Voronoi analysis, where each Voronoi vertex becomes the potential interstitial site. Subsequently, a vacuum layer with the width of over 10 Å was inserted into at the interface of two grain boundary models, that is, two surfaces are nucleated. The cleavage energy is defined as the energy difference between grain boundary and surface models. This energy would provide important indicator to predict the effect of solutes on fracture process. It was confirmed that these two grain boundaries have similar tendency for the segregation.

**First-principles calculations.** First-principles calculations were carried out within the density functional theory (DFT) framework using the Vienna ab initio simulation package (VASP)[25,26]. Projector augmented wave potentials[27] were employed with the Perdew–Burke–Ernzerhof generalized gradient approximation exchange-correlation density functional[28]. The Brillouin-zone gamma-centered $k$-point samplings were chosen using the Monkhorst–Pack algorithm[29], where a $3 \times 3 \times 3$, $4 \times 8 \times 3$, and $3 \times 8 \times 1$ grid was used for perfect crystal, $(10\bar{1}4)$ and $(30\bar{3}2)$ grain boundaries, respectively. A cut-off in plane-wave energy of 500 eV was applied using a first-order Methfessel–Paxton scheme that employed a smearing parameter of 0.2 eV. The total energy was converged within $10^{-6}$ eV/atom for all calculations. The relaxed configurations were obtained using the conjugate gradient method until the force on all atoms was reduced to 0.01 eV/Å. Atomic configurations were visualized using VESTA[30].

**Theory of ideal fracture.** To evaluate the influence of solutes on the fracture behavior, an energy-based Griffith criterion for crack propagation was applied. The energy release rate under plane strain condition can be expressed as $(1 - \nu^2)K_{IC}^2/E = 2\gamma_s$, where $K_{IC}$ is the critical stress intensity factor, $\nu$ is Poisson's ratio, $E$ is the Young's modulus, and $\gamma_s$ is the surface energy per area. In metals exhibiting ductile fracture behavior, the contribution of plastic work needs to be added to the energy balance[31–33]: $(1 - \nu^2)K_{IC}^2/E = 2\gamma_s + \gamma_p$, where the plastic work term $\gamma_p$ can be associated with the surface energy $\gamma_p(\gamma_s)$ as $(1 - \nu^2)K_{IC}^2/E = 2\gamma_s + \gamma_p(\gamma_s)$[34]. Accordingly, $\gamma_s$ can be a unique indicator of the fracture toughness. For interfacial fracture, such as grain boundary (GB) fracture, the excess energy of the interface $\gamma_{GB}$ is subtracted from the surface energy and the ideal work of interfacial

cleavage $2\gamma_{int}$ is expressed as $2\gamma_s \Rightarrow 2\gamma_s - \gamma_{GB} \equiv 2\gamma_{int}$. Rice and Wang[35] discussed embrittlement of interfaces by solution segregation using $2\gamma_{int}$. Because the change of the surface and interfacial energies corresponds to the segregation energy at the surface and interface, the ideal work of interfacial cleavage with solution segregation $2\gamma_{int}^{seg}$ can be expressed as $2\gamma_{int}^{seg} = (2\gamma_s + \Delta E_s^{seg}\Gamma) - (\gamma_{TB} + \Delta E_{GB}^{seg}\Gamma)$, where $\Delta E_s^{seg}$ and $\Delta E_{GB}^{seg}$ are change in segregation energy at surface and interface of a solute, $\Gamma$ is a conversion factor of segregation energy into energy per area.

Accordingly, the difference in the ideal work of interfacial cleavage between pure metals and alloys ($\Delta 2\gamma_{int}^{seg} = 2\gamma_{int}^{seg} - 2\gamma_{int} = (\Delta E_s^{seg} - \Delta E_{GB}^{seg})\Gamma$) can be a reliable indicator to predict the effect of grain boundary solute segregation on the interfacial fracture behavior, i.e. causing either cohesion or decohesion tendency. In the present study, DFT calculations were performed to evaluate these predictive factors directly. The periodic boundary condition was applied to the direction normal to the GB plane, and the energy difference between two configurations, that is grain boundary segregation and the surface absorption states, were calculated using atomic models (Supplementary Fig. S11). It was reported that this prediction is well correlated with the macroscopic fracture toughness[36–38].

## Data availability
All data needed to evaluate the conclusions in the paper are present in the paper and/or the Supplementary Materials. Additional data related to this paper may be requested from the authors.

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

## Acknowledgements

The authors on Japan side would like to acknowledge the financial support from Elements Strategy Initiative for Structural Materials (ESISM, No. JPMXP0112101000) in Kyoto University, JST CREST (JPMJCR1994), and the Grant-in-Aid for Scientific Research (No. JP15H05767, JP18H05455, JP18H05456 and JP20H00306) supported by the Ministry of Education, Culture, Sports, Science and Technology (MEXT), Japan. T.T. acknowledges the support of JST PRESTO Grant Number JPMJPR1998 and JSPS KAKENHI (Grant Numbers. JP19K04993). Simulations were performed on the large-scale parallel computer system with HPE SGI 8600 at JAEA. N.T. appreciates to the support from the Light Metal Educational Foundation, Japan. J.W.M and A.M.M are grateful to the funding from the US Office of Naval Research under Grant No. N00014-19-1-2376. Work at the Molecular Foundry was supported by the Office of Science, Office of Basic Energy Sciences, of the U.S. Department of Energy under Contract No. DE-AC02-05CH11231.The support on TEM experiments from Kenji Kazumi and Nobuharu Sasaki of Kyoto University is also appreciated.

## Author contributions

Y.C., A.M.M. and N.T. prepared the manuscript, which was reviewed and edited by all authors; Y.C. conducted the HPT and annealing experiments; Y.C. conducted the SEM-BSE characterization; Y.C. conducted the EBSD characterization; Y.C. conducted the tensile tests and data analysis; Y.C. and R.Z. conducted the TEM experiments; Y.C., W.G. and A.G. conducted the HR-EBSD experiments and strain field analysis; M.M. conducted the dislocation tomography analysis; Y.C., R.G. and K.I. conducted the FIB and APT experiments, as well as data processing and analysis; T.T. conducted the theoretical calculations on GB segregation of oxygen and its effect on GB separation energy. Project administration, supervision, and funding acquisition were performed by J.W.M., A.M.M and N.T.

## Competing interests

The authors declare no competing interests.
