## [Peer Review File · Nature Communications]

Grain refinement in titanium prevents low temperature oxygen embrittlementREVIEWER COMMENTS

Reviewer #1 (Remarks to the Author):

Key results

In this work, the authors studied the influence of grain size on the tensile behavior of alpha-Ti with and without oxygen (Ti and Ti-0.3O, or 0.3 wt% O). The results reveal an increase in ductility with decreasing grain size in Ti-0.3O alloy, which is attributable to twinning (or lack thereof), the occurrence of nanotwinning to relieve stress concentrations at twin/grain boundaries intersections, and dilution of O segregation when the grain size becomes small (ultra-fine grained, UFG). The latter effect occurs in concert with Fe segregation at grain boundaries which also draws the O to those regions. The lack of planar slip and a more homogeneous distribution of dislocations associated with activation of c+a dislocation glide in addition to a glide in the UFG Ti-0.3O alloy differs from the CG Ti-0.3O or UFG Ti and means the UFG Ti-0.3O is better able to maintain deformation compatibility because there are more active slip systems in this material, contributing to its superior ductility.

Validity

The interpretation of the data is appropriate and well-supported. In addition, the Conclusions are generally supported by the data, except on lines 311-316, the conclusion that segregation of other trace elements to grain boundaries could also benefit tensile ductility in UFG materials is speculative and not directly supported by measurements or modeling. It would be more appropriate to say that the segregation of other elements to grain boundaries should be studied to determine whether they are more effective than oxygen in improving ductility.

Significance

This paper presents significant new findings on a way to improve the ductility of Ti in oxygen-containing environments through grain boundary engineering. It is of both scientific and engineering interest.

Data and methodology

The quality of the experimental data is high, and the methods used to analyze the results are appropriate. The authors provide good experimental details in the Methods and Materials section. Minor concerns are noted below under Suggested Improvements. The reviewer is not experienced in DFT calculation methods, hence is not able to assess the quality of the modeling methods described in Computational Methods sections (i) and (ii) in detail. In addition, the format of the supplementary videos was unknown and the review wasn't able to view them.

Analytical approach

As noted under Data and Methodology, the analytical approaches to analyzing the data are appropriate.

Suggested improvements

There are several aspects that could be improved to enhance the quality and impact of the paper. These include:

It would be helpful to mention that the alloy is HCP alpha-Ti earlier in the paper (in the Introduction). That's now buried in the Materials and Methods".

Line 49 - The authors should note that 2 microns is on the upper limit of what is considered to be ultra-fine grained (UFG).

Figure 1(a) – The criterion used to separate high angle from low angle grain boundaries should be specified. 10 degrees misorientation?

Line 70 – the word “drag-on” should just be “drag”.

Line 74 – The word “were” should be “was”.

Line 87 – The word “dimple” should be “dimpled”.

Lines 296-297 - The phrase “propagation can be shortly stopped” should be “propagation can be stopped after propagating a short distance” to clarify the meaning.

Line 334 - The specimen thickness is 0.6 mm or 600 microns, so for a grain size of 164 microns in the Ti they don't have true polycrystalline behavior (only 3-4 grains through the thickness) and for 68 microns in the Ti-0.30 there is just barely polycrystalline behavior (9 grains through the thickness). The authors must address this issue to ensure that the reader understands it, even though there appears to be no obvious impact on the mechanical properties.

Line 345 – The word “reproductivity” should be “reproducibility”.

Line 349 – The proper name is “Kroll's reagent”; also Line 391

Line 356 – The correct term is “line intercept method”.

Line 365 – The word “invisible” should be “invisibility”.

Line 411 - This sentence should refer to Fig. S4b and d rather than Figure S5b and d.

Line 448 - The phrase “were inserted” should be “was inserted”.

Lines 472-490 - The stress intensity factor should be KI, not KIc. Using the subscript “IC” implies it represents the fracture toughness under plane strain conditions. Furthermore, it is not appropriate to use a local stress (σ^*) to calculate the stress intensity factor. KI is a global (not local) parameter that depends on geometry, applied stress and crack size. If the authors mean that they are considering crack propagation in the grain boundary, they should state that. However, it is very difficult to properly account for a localized fracture event while maintaining equilibrium and compatibility with the surrounding material. There is no discussion of boundary conditions in this section, so it is not clear exactly what they have done. Significant clarification of the fracture modeling is needed.

Line 754 - The word “tensiled” should be “tensile”.

In some places the authors use proper crystallographic notation with bars over indices and in other places they use negative signs. It would be good to be consistent and to use the proper bars.

Clarity and context

The text is well-organized, and the meaning is clear throughout other than as noted above. The figures in the main text are all multi-part and it is not clear to the reviewer that this benefits the presentation. It seems that it would be clearer to put items that are not directly related to one another (especially in Figures 1-3) in separate figures.

References

The references are appropriate and current.

Reviewer #2 (Remarks to the Author):

It has long been known that titanium can form a solid solution with oxygen, leading to embrittlement; for more information, see the extensive published literature on 'alpha case.' Alpha case embrittlement is characterized by a simultaneous decrease in ductility and increase in strength due to the effects of oxygen in inhibiting dislocation motion under applied stresses. Prior work in a number of other alloy systems has shown that severe plastic deformation can be used to produce ultra-fine grain sizes and corresponding increases in yield strength. Further, prior work has also shown that, at a certain fineness of grain size, there is a tipping point wherein the loss of ductility typical of increases in strength is reversed, and ductility again rises with further grain size refinement. To my knowledge, this is the first work showing that this effect also occurs in Ti with a substantial amount of dissolved O, known to cause severe embrittlement. The magnitude of the ductility improvement after grain refinement is so great that the ductility exhibited is sufficient for engineering of structures requiring damage tolerance. If such processing could be performed on an appropriate bulk scale, it could lead to new possibilities for spacecraft and space access vehicles.

The manuscript is well written and the work supports the authors' conclusions. Minor revisions are recommended to address the designation of a CSL boundary on line 195: Insufficient information appears to be given, as a CSL boundary is usually described by a Sigma value related to the fraction of coincident sites in perfect lattices. The boundary cannot be described only in terms of a plane because there must be a rotation between the lattices. The assumption that the Young's modulus is roughly the same between cryogenic and room temperatures should also be examined further, to further verify the significance of the findings. What difference in modulus should be considered insignificant in the present work? Modulus is known to be a function of temperature and is likely to vary by ~1% (or possibly more) in this temperature regime. The atom probe tomography results show segregation of Fe in addition to O at the grain boundary. What role does Fe content play in O presence at the grain boundary? It is interesting that segregation of these elements is found in the ultra-fine grained specimen because recrystallization should result in the grain boundaries being in different locations than during the original processing that produced the coarse grained material. This raises the question of what temperatures were reached during high pressure torsion and how long did the material remain at an elevated temperature such that the Fe and O atoms could diffuse to the grain boundaries.

Reviewer #3 (Remarks to the Author):

Chong et al. report in this work a new strategy to alleviate the oxygen induced embrittlement in Ti at low temperature. The uniform elongation of an ultrafine-grained (UFG) microstructure in Ti-0.3wt.%O was successfully increased by an order of magnitude, while maintaining an ultrahigh yield strength inherent to the UFG microstructure. Grain refinement proves to be a very efficient approach to fight

against oxygen embrittlement. This finding is of huge engineering significance and deserves publication in Nature Communications. Regarding the part of first-principles calculations and the discussions, however, I do have some concerns which deter me from recommending publication of this manuscript in its current form.

(1) The reported calculations show that Fe-O interaction is slightly repulsive (0.03 eV) in bulk Ti, but strongly attractive (1.20 eV) at the high-energy grain boundaries. What is the physics underlying this huge difference? If the Fe-O bonding will be strongly enhanced in the presence of free volume, should it be also the case near the vacancy in the bulk? I had my student have a preliminary test calculation on the latter case just now, and it is found the Fe-vacancy pair only shows very weak attraction (<0.1 eV) to O. Therefore, the reported extremely strong Fe-O attraction at the grain boundary is doubtful to me, to say the least.

(2) Knowing Fe-O interaction is so strong at the grain boundary, the authors still calculating only the separation energy of O in the absence of Fe. It does not make sense to me.

(3) When the authors cleave the crystal along the grain boundary by inserting a vacuum, they leave all O atoms in one half and generate a clean Ti surface for the other. Maybe this is energetically reasonable, but should be justified carefully.

(4) The energy cutoff employed in the calculations was 450 eV. I would suggest 500 eV for O, for a high-quality result.

(5) The term “separation energy” is somewhat misleading. It was supposed to mean the effect of an impurity or alloying element on the cleavage energy along the grain boundary, but its literal meaning to many readers in this field should be the cleavage energy. So, what the authors really mean by this term is the change of separation energy.

(6) In Fig. 4h, left panel, segregation energy of an impurity is presented in unit of eV/atom, but in the right panel the effect on cleavage energy is in unit J/m^2 . It is better to use the same unit for readers' convenience in case they want to compare these two quantities.

REVIEWER COMMENTS

Reviewer #1 (Remarks to the Author):

Key results

In this work, the authors studied the influence of grain size on the tensile behavior of alpha-Ti with and without oxygen (Ti and Ti-0.3O, or 0.3 wt% O). The results reveal an increase in ductility with decreasing grain size in Ti-0.3O alloy, which is attributable to twinning (or lack thereof), the occurrence of nanotwinning to relieve stress concentrations at twin/grain boundaries intersections, and dilution of O segregation when the grain size becomes small (ultra-fine grained, UFG). The latter effect occurs in concert with Fe segregation at grain boundaries which also draws the O to those regions. The lack of planar slip and a more homogeneous distribution of dislocations associated with activation of c+a dislocation glide in addition to a glide in the UFG Ti-0.3O alloy differs from the CG Ti-0.3O or UFG Ti and means the UFG Ti-0.3O is better able to maintain deformation compatibility because there are more active slip systems in this material, contributing to its superior ductility.

Validity

The interpretation of the data is appropriate and well-supported. In addition, the Conclusions are generally supported by the data, except on lines 311-316, the conclusion that segregation of other trace elements to grain boundaries could also benefit tensile ductility in UFG materials is speculative and not directly supported by measurements or modelling. It would be more appropriate to say that the segregation of other elements to grain boundaries should be studied to determine whether they are more effective than oxygen in improving ductility.

Response to reviewer's comment: We appreciate the reviewer for this nice comment. We agree with the reviewer that whether a segregation of other trace elements to the grain boundaries will benefit or even deteriorate the tensile ductility of Ti-O alloy is unknown, which would require an additional detailed study on this specific point. In this regard, we modified the relevant descriptions in the conclusion of the revised manuscript according to the reviewer's suggestion, as follows:

(Page 13-14)

'Moreover, in light of the harmful effect of oxygen segregation on the grain boundary cohesive energy and tensile ductility, a potential grain boundary chemistry engineering approach can be proposed for Ti-O alloys. By doping the alloy with other trace elements (e.g. boron and yttrium) that have a stronger grain boundary segregation tendency than oxygen, the grain boundary cohesivity of Ti-O alloys could be either enhanced or further decreased, leading to an improved or even deteriorated tensile ductility of the material. This, however, requires further experimental and modelling work with respect to each specific doping element.'

Significance

This paper presents significant new findings on a way to improve the ductility of Ti in oxygen-containing environments through grain boundary engineering. It is of both scientific and engineering interest.

Data and methodology

The quality of the experimental data is high, and the methods used to analyze the results are appropriate. The authors provide good experimental details in the Methods and Materials section. Minor concerns are noted below under Suggested Improvements. The reviewer is not experienced in DFT calculation methods, hence is not able to assess the quality of the modelling methods described in Computational Methods sections (i) and (ii) in detail. In addition, the format of the supplementary videos was unknown and the review wasn't able to view them.

Response to reviewer's comment: We appreciate the reviewer for pointing out this issue and apologize for using an unconventional format for our videos. We have changed the format of the supplementary videos (3D dislocation tomography) into the '.avi' version (a more common type) in the revised manuscript. We hope they can be accessible to the reviewer this time.

Analytical approach

As noted under Data and Methodology, the analytical approaches to analyzing the data are appropriate.

Suggested improvements

There are several aspects that could be improved to enhance the quality and impact of the paper. These include:

It would be helpful to mention that the alloy is HCP alpha-Ti earlier in the paper (in the Introduction). That's now buried in the Materials and Methods".

Response to reviewer's comment: This is a good point! In the revised manuscript, we mentioned this information earlier in the first and second paragraphs of the Introduction part, as follows:

(Page 2)

'Interstitial oxygen has long been considered as a double-edged sword in pure titanium with a hexagonal close packed (hcp) structure.'

'Here, we propose a fundamental strategy to overcome this dilemma in hcp-titanium via grain refinement.'

Line 49 - The authors should note that 2 microns is on the upper limit of what is considered to be ultra-fine grained (UFG).

Response to reviewer's comment: We appreciate the reviewer for this nice comment. We have added the suggested clarifications in the revised manuscript to avoid misleading to the readers. The revised description in the manuscript is as follows:

(Page 2)

'It is demonstrated in ultrafine-grained (UFG) (average grain sizes $D \sim 2.0 \mu\text{m}$, close to the upper limit of what can be considered as UFG structure) titanium containing 0.3 wt.% oxygen that...'

Figure 1(a) – The criterion used to separate high angle from low angle grain boundaries should be specified. 10 degrees misorientation?

Response to reviewer's comment: We appreciate the reviewer for this nice comment and apologize for not mentioning the criterion in the previous version. The high angle grain boundaries were termed as boundaries

with misorientation angles, $\theta > 15^\circ$, while low angle grain boundaries were termed as those with $2^\circ < \theta \leq 15^\circ$. We have also added relevant descriptions in the figure caption of Fig. 1 as well as in the main text as follows:

(Page 3)

'Representative grain boundary maps (blue lines: high-angle grain boundaries with misorientation angle, $\theta > 15^\circ$; red lines: low-angle grain boundaries with misorientation angle, $2^\circ < \theta \leq 15^\circ$) of pure Ti and Ti-0.3O...'

Line 70 – the word “drag-on” should just be “drag”.

Line 74 – The word “were” should be “was”.

Line 87 – The word “dimple” should be “dimpled”.

Lines 296-297 - The phrase “propagation can be shortly stopped” should be “propagation can be stopped after propagating a short distance” to clarify the meaning.

Response to reviewer's comment: We appreciate the reviewer for pointing these grammar and language issues of our manuscript. We have modified them in the revised manuscript accordingly.

Line 334 - The specimen thickness is 0.6 mm or 600 microns, so for a grain size of 164 microns in the Ti they don't have true polycrystalline behavior (only 3-4 grains through the thickness) and for 68 microns in the Ti-0.3O there is just barely polycrystalline behavior (9 grains through the thickness). The authors must address this issue to ensure that the reader understands it, even though there appears to be no obvious impact on the mechanical properties.

Response to reviewer's comment: This is a nice point! Indeed, there are only limited number of grains in the thickness direction of the coarse-grained samples for both alloys. This could lead to uncertainty for the tensile properties obtained from these CG samples, which can hardly be considered as polycrystalline microstructures. To this end, we prepared some larger-sized tensile specimens (gauge geometry: 10 mm (l) * 2 mm (w) * 1.15 mm (t)) for the CG microstructures (where severe plastic deformation is not necessary) of both alloys that contained more grains in the thickness direction. We tested their tensile properties at 77 K and compared them with those obtained from micro-sized tensile specimens. The result is shown in Fig. A, below. It agreed well with what the reviewer suggested that, the tensile stress-strain curves were quite similar between specimens with different thickness, i.e., there is no obvious impact of the sample thickness on the mechanical properties.

Nonetheless, we agree with the reviewer that the small specimen thickness of CG microstructures may lead to misunderstanding to the readers. So, we added Fig. A as supplementary Fig. S10 in the revised manuscript. We also added relevant descriptions in the Experimental Methods part in the revised manuscript as follows:

(Page 15)

'Due to the limited thickness (~0.6 mm) of the tensile specimens, only a few grains were included in the thickness direction of coarse-grained microstructures of pure Ti ($D = 164 \mu\text{m}$) and Ti-0.3O ($D = 68 \mu\text{m}$) alloys, which may cause uncertainties in the obtained mechanical properties, as they can hardly be considered as polycrystalline specimens. To this end, we also prepared some larger-sized tensile specimens (gauge geometry: 10 mm (l) \times 2 mm (w) \times 1.15 mm (t)) for the CG microstructures (where severe plastic deformation is not necessary) of both alloys, and compared their tensile properties at 77 K with those from micro-sized tensile specimens. The comparison is shown in Fig. S10. It is clearly shown that the engineering stress-strain curves of

specimens with different thickness were quite similar for both pure Ti and Ti-0.3O alloys with a coarse grain size. This confirms the reliability of the mechanical properties we obtained for CG microstructures from the micro-sized tensile specimens.'

Fig. A Validation of the sample thickness effect on the tensile properties of coarse-grained pure Ti ($D = 164 \mu\text{m}$) and Ti-0.3O ($D = 68 \mu\text{m}$) alloys at 77 K. A good consistency was found for the tensile properties obtained from samples with different thickness of both alloys.

Line 345 – The word “reproductivity” should be “reproducibility”.

Line 349 – The proper name is “Kroll’s reagent”; also Line 391

Line 356 – The correct term is “line intercept method”.

Line 365 – The word “invisible” should be “invisibility”.

Line 411 - This sentence should refer to Fig. S4b and d rather than Figure S5b and d.

Line 448 - The phrase “were inserted” should be “was inserted”.

Response to reviewer’s comment: We appreciate the reviewer’s detailed corrections of our manuscript. They are truly helpful and we have corrected them in the revised manuscript accordingly.

Lines 472-490 - The stress intensity factor should be K_I , not k_{IC} . Using the subscript “IC” implies it represents the fracture toughness under plane strain conditions. Furthermore, it is not appropriate to use a local stress (σ^*) to calculate the stress intensity factor. K_I is a global (not local) parameter that depends on geometry, applied stress and crack size. If the authors mean that they are considering crack propagation in the grain boundary, they should state that. However, it is very difficult to properly account for a localized fracture event while maintaining equilibrium and compatibility with the surrounding material. There is no discussion of boundary conditions in this section, so it is not clear exactly what they have done. Significant clarification of the fracture modelling is needed.

Response to reviewer’s comment: We appreciate the reviewer for this nice comment. As the reviewers suggested, we should have used K_I instead of k_{IC} , because we do not intend to discuss the local event for the crack propagation in the grain boundary. In the present study, we discuss the ideal value of the fracture

toughness without any local geometry and condition. That corresponds to the upper limit of the toughness. In that sense, we agree with the reviewer’s comment: the value should be defined as a global parameter (so we should use K (capital) instead of k (lowercase)). On the other hand, we think that “ K_{IC} ” can be used because this parameter corresponds to the critical value of fracture toughness for ideal cleavage. In this regard, we removed and modified related descriptions in the revised manuscript as shown below.

In addition, as the reviewer also pointed out, it is very difficult to properly account for a localized fracture event directly by the DFT calculation. Instead of the realistic crack propagation model, we considered the energy-based criterion for the fracture. Here, we applied the periodic boundary condition to the direction normal to the grain boundary and evaluated the ideal value based on the energy difference between the interface and the surface. According to the reviewers’ suggestion, we added detailed information for the calculation condition in the Experimental Method part as follows:

(Page 18-19)

‘The energy-based Griffith criterion for crack propagation was applied to evaluate the effect of solutes. The energy release rate under plane strain condition can be expressed as $(1 - \nu^2)K_{IC}^2/E = 2\gamma_s$, where K_{IC} is the critical stress intensity factor, ν is Poisson’s ratio, E is the Young’s modulus, and γ_s is the surface energy per area. In ductile fracture of metals, the contribution of plastic work needs to be added to the energy balance [31–33]: $(1 - \nu^2)K_{IC}^2/E = 2\gamma_s + \gamma_p$, where the plastic work term γ_p can be expressed as a function of the surface energy $\gamma_p(\gamma_s)$ [34]: $(1 - \nu^2)K_{IC}^2/E = 2\gamma_s + \gamma_p(\gamma_s)$. Accordingly, γ_s is a unique property describing the fracture toughness. In the case of interfacial fracture, such as a grain boundary (GB), the excess energy of the interface γ_{GB} is subtracted from the surface energy and the ideal work of interfacial cleavage $2\gamma_{int}$ is expressed as $2\gamma_s \Rightarrow 2\gamma_s - \gamma_{GB} \equiv 2\gamma_{int}$. Rice and Wang discussed embrittlement of interfaces by solution segregation using $2\gamma_{int}$ [35]. Because the change of the surface and interfacial energies corresponds to the segregation energy at the surface and interface, the ideal work of interfacial cleavage with solution segregation $2\gamma_{int}^{seg}$ can be expressed as $2\gamma_{int}^{seg} = (2\gamma_s + \Delta E_s^{seg} \Gamma) - (\gamma_{TB} + \Delta E_{GB}^{seg} \Gamma)$, where ΔE_s^{seg} and ΔE_{GB}^{seg} are change in segregation energy at surface and interface of a solute, Γ is a conversion factor of segregation energy into energy per area.

Accordingly, the difference in the ideal work of interfacial cleavage between alloys and pure metals ($\Delta 2\gamma_{int}^{seg} = 2\gamma_{int}^{seg} - 2\gamma_{int} = (\Delta E_s^{seg} - \Delta E_{GB}^{seg})\Gamma$) can be regarded as the dominant factor for interfacial fracture that predicts the cohesion and decohesion tendencies with addition of solutes. In the present study, DFT calculations were performed to evaluate these predictive factors directly. The periodic boundary condition was applied to the direction normal to the GB plane, and the energy difference between two configurations, that is grain boundary segregation and the surface absorption states, were calculated using atomic models (Fig. S11). It was reported that this prediction is well correlated with the macroscopic fracture toughness [36–38].’

Line 754 - The word “tensiled” should be “tensile”.

In some places the authors use proper crystallographic notation with bars over indices and in other places they use negative signs. It would be good to be consistent and to use the proper bars.

Response to reviewer's comment: We appreciate the reviewer's detailed corrections of our manuscript. We have corrected the typo, and also changed all the crystallographic notations using bars over indices. They are highlighted by yellow colors in the revised manuscript.

Clarity and context

The text is well-organized, and the meaning is clear throughout other than as noted above. The figures in the main text are all multi-part and it is not clear to the reviewer that this benefits the presentation. It seems that it would be clearer to put items that are not directly related to one another (especially in Figures 1-3) in separate figures.

Response to reviewer's comment: We appreciate the reviewer's comment on this issue. Indeed, Figures 1-3 are somehow crowded, composed of several images. In the revised manuscript, we divided Figure 1 into two separate figures, one showing the GB-maps of pure Ti and Ti-0.3O alloys with different grain sizes, the other illustrating the mechanical properties and fracture surfaces of these microstructures.

However, as for Figures 2 and 3, we would like to keep them as the original formats, if possible. The two figures served as a direct comparison of the mesoscopic deformation behaviors between CG pure Ti and Ti-0.3O (Figure 2), as well as UFG pure Ti and Ti-0.3O alloys (Figure 3). If they are divided into several separate figures, it might cause some difficulties for the readers interpreting the difference in the deformation microstructures between the two alloys. We truly appreciate the reviewer's kind understanding on this issue.

References

The references are appropriate and current.

Reviewer #2 (Remarks to the Author):

It has long been known that titanium can form a solid solution with oxygen, leading to embrittlement; for more information, see the extensive published literature on 'alpha case.' Alpha case embrittlement is characterized by a simultaneous decrease in ductility and increase in strength due to the effects of oxygen in inhibiting dislocation motion under applied stresses. Prior work in a number of other alloy systems has shown that severe plastic deformation can be used to produce ultra-fine grain sizes and corresponding increases in yield strength. Further, prior work has also shown that, at a certain fineness of grain size, there is a tipping point wherein the loss of ductility typical of increases in strength is reversed, and ductility again rises with further grain size refinement. To my knowledge, this is the first work showing that this effect also occurs in Ti with a substantial amount of dissolved O, known to cause severe embrittlement. The magnitude of the ductility improvement after grain refinement is so great that the ductility exhibited is sufficient for engineering of structures requiring damage tolerance. If such processing could be performed an appropriate bulk scale, it could lead to new possibilities for spacecraft and space access vehicles.

Response to reviewer's comment: We truly appreciate the reviewer's positive comments on our results of UFG Ti-O alloys. In addition, it is really a good point to consider applying the grain refinement strategy to bulk scale, opening new possibilities for spacecraft and space access vehicles.

In this study, the minimum grain size obtained for Ti-0.3O alloy is 2.0 μm , which has been proved to be sufficient to eliminate the low temperature embrittlement (caused by oxygen atoms), as typically observed in the coarse-grained counterpart. Such a grain size, however, is not very small (at the upper limit of what can be called ultrafine-grained microstructure), which is achievable by conventional cold rolling and annealing process, at least for high-purity titanium. We have experiences in obtaining ultrafine-grained commercial purity (CP) Ti (grade 2), with a grain size of $\sim 1.0 \mu\text{m}$ by heavy cold rolling (a total thickness reduction of 92%) with several moderate intermediate annealing between cold rolling passes. However, we haven't tried the same procedure in Ti-0.3O alloy (similar to the impurity content of CP-Ti, grade 4). Difficulty might be expected for cold rolling of Ti-0.3O alloy to the same thickness reduction as in CP-Ti (grade 2), due to the greatly increased strength and deteriorated ductility/workability with increasing oxygen content. Nevertheless, it is believed that, through a careful control of the cold reduction and the intermediate annealing between each rolling pass, it is possible to achieve a similar grain size as we reported here using the HPT and annealing method. This is surely an interesting topic that we will further dig into in the near future.

The manuscript is well written and the work supports the authors' conclusions. Minor revisions are recommended to address the designation of a CSL boundary on line 195: Insufficient information appears to be given, as a CSL boundary is usually described by a Sigma value related to the fraction of coincident sites in perfect lattices. The boundary cannot be described only in terms of a plane because there must be a rotation between the lattices.

Response to reviewer's comment: We appreciate the reviewer for pointing out this misleading term in our manuscript. As the reviewer pointed out, the type of CSL grain boundary cannot be identified uniquely by only the information of GB plane. The GB structures of $(10\bar{1}4)$ and $(30\bar{3}2)$ boundaries used in this study correspond

to the $\Sigma 22(11)$ and $\Sigma 27$ CSL boundaries with a $\langle 11\bar{2}0 \rangle$ rotation axis, respectively, assuming an ideal c/a ratio. We have also added relevant descriptions in the revised manuscript as follows:

(Page 8)

'The $(10\bar{1}4)$ coincidence site lattice (CSL) grain boundary corresponding near $\Sigma 22$ with a $\langle 11\bar{2}0 \rangle$ rotation axis, which are energetically unstable compared to the other twin and grain boundaries from preliminary calculations, was taken as suitable examples of HAGB.'

The assumption that the Young's modulus is roughly the same between cryogenic and room temperatures should also be examined further, to further verify the significance of the findings. What difference in modulus should be considered insignificant in the present work? Modulus is known to be a function of temperature and is likely to vary by $\sim 1\%$ (or possibly more) in this temperature regime.

Response to reviewer's comment: We truly appreciate the reviewer for this nice comment. We totally agree with the reviewer that the Young's modulus is temperature dependent and increases with decreasing temperature (as the reviewer suggested, the variation could be more than 1%).

In this study, we obtained an accurate Young's modulus of the specimens at room temperature, by using digital image correlation (DIC) method to precisely measure the elastic region during tensile deformation. The tensile specimen surface was sprayed with white and black contrast particles. We took videos of the whole tensile deformation process, and then calculated the accurate strain by tracking the locations of those particles during tensile deformation, using commercial Vic-2D software. This is, however, technically difficult (or maybe impossible) for tensile deformation at liquid nitrogen temperature (LN_2), as both the tensile specimens and jigs were fully immersed in the LN_2 . In this regard, we were not able to experimentally measure the accurate Young's modulus of the specimens at LN_2 temperature, at the current stage to say at least. That is the reason why we assume a roughly same Young's modulus between the two testing temperatures in the manuscript.

Although there exists a slight difference in Young's modulus between the two testing temperatures, the variation is comparatively small compared with the dramatic changes in tensile ductility caused by deformation temperature, oxygen content and grain size. Nevertheless, we truly appreciate the reviewer for bringing out this issue, and accordingly, we modified the relevant descriptions in the revised manuscript as follows:

(Page 14-15)

'The displacements obtained from tensile machine were calibrated using the room temperature strain data (obtained by the DIC method), assuming that the Young's modulus of the material was roughly the same at two testing temperatures. It should be noted that the Young's modulus actually slightly increased with decreasing temperature (in some cases, the variations could be around 1%). However, this variation is comparatively smaller than the changes of total elongation caused by different deformation temperatures, grain sizes and oxygen contents, as seen in this study.'

The atom probe tomography results show segregation of Fe in addition to O at the grain boundary. What role does Fe content play in O presence at the grain boundary? It is interesting that segregation of these elements is found in the ultrafine-grained specimen because recrystallization should result in the grain boundaries being in different locations than during the original processing that produced the coarse grained material. This raises

the question of what temperatures were reached during high pressure torsion and how long did the material remain at an elevated temperature such that the Fe and O atoms could diffuse to the grain boundaries.

Response to reviewer's comment: We appreciate the reviewer for this comment. Indeed, there is also a strong segregation of Fe at the grain boundaries, especially for the coarse-grained Ti-0.3O microstructure, as shown by the APT results in Supplementary Figs. S4 and S5.

In this study, we employed first-principles calculations to study the role of Fe atoms played in the grain boundary segregation behavior of O atoms. The $(10\bar{1}4)$ grain boundary (close to $\Sigma 22$ boundary with $\langle 11\bar{2}0 \rangle$ rotation axis) was taken as a representative example for HAGBs in hcp-Ti. The segregation energies of oxygen atoms *alone* at the grain boundary, as well as oxygen atoms *together* with iron atoms at the grain boundary were calculated (shown in the segregation panel of Fig. 5h). It is interesting to see a high positive segregation energy (0.6 eV, bar with an orange color in the segregation panel of Fig. 5h) for oxygen atoms staying at the grain boundary *alone*. This result indicates that oxygen atoms segregating at the grain boundaries *alone* are not energetically favored in hcp-Ti. However, it is striking to see that the segregation energy became negative when oxygen and iron atoms are staying *together* at the grain boundary (-0.6 eV, bar with a purple color in the segregation panel of Fig. 5h). This suggests that the co-segregation of oxygen and iron atoms *together* at the grain boundaries is energetically favored in hcp-Ti. Such an attractive tendency between oxygen and iron atoms at the grain boundary thus provides a reasonable explanation for the experimental observed co-segregation of iron and oxygen atoms together at the grain boundaries of CG Ti-0.3O alloy. That is to say, iron atoms at the grain boundaries probably attract oxygen atoms to the grain boundaries.

The iron segregation at grain boundary was still present in the UFG Ti-0.3O microstructure, but the segregation level has been substantially lowered, as shown by the elemental profiles shown in Fig. B (obtained by APT data analysis). This can explain the relatively homogenous distribution of oxygen atoms across the grain boundaries in the UFG Ti-0.3O microstructure (Fig. 5g).

The temperate rise during HPT deformation (at a speed of 0.5 rotation per minute) of Ti-O alloys can be roughly estimated to be 50~100°C. In this regard, we believe that the GB-segregation of oxygen and iron atoms occurs at the annealing process (480~880°C) after HPT deformation, and the annealing time (600 s) is sufficient to reach an equilibrium state of GB-segregation. The GB-segregation of oxygen and iron atoms achieved at the annealing temperature can be maintained after the subsequent water quenching, and then plays a critical role in affecting the deformation behaviors of the materials.

Fig. B The elemental profiles of iron atoms across the grain boundaries in coarse-grained (CG) and ultrafine-grained (UFG) Ti-0.3O alloy. The segregation level of iron atoms at the grain boundary has been substantially lowered by grain refinement.

Reviewer #3 (Remarks to the Author):

Chong et al. report in this work a new strategy to alleviate the oxygen induced embrittlement in Ti at low temperature. The uniform elongation of an ultrafine-grained (UFG) microstructure in Ti-0.3wt.%O was successfully increased by an order of magnitude, while maintaining an ultrahigh yield strength inherent to the UFG microstructure. Grain refinement proves to be a very efficient approach to fight against oxygen embrittlement. This finding is of huge engineering significance and deserves publication in Nature Communications. Regarding the part of first-principles calculations and the discussions, however, I do have some concerns which deter me from recommending publication of this manuscript in its current form.

(1) The reported calculations show that Fe-O interaction is slightly repulsive (0.03 eV) in bulk Ti, but strongly attractive (1.20 eV) at the high-energy grain boundaries. What is the physics underlying this huge difference? If the Fe-O bonding will be strongly enhanced in the presence of free volume, should it be also the case near the vacancy in the bulk? I had my student have a preliminary test calculation on the latter case just now, and it is found the Fe-vacancy pair only shows very weak attraction (<0.1 eV) to O. Therefore, the reported extremely strong Fe-O attraction at the grain boundary is doubtful to me, to say the least.

Response to reviewer's comment (1): We truly appreciate the reviewer for this nice comment, which is also a strong concern to us. Indeed, we have also performed the same calculation as the reviewer did, i.e. calculating the binding energy of Fe-vacancy pair with oxygen in the bulk to investigate the effect of the free volume. Then we obtained the similar results as the reviewer did. However, there is no doubt from our experiments that oxygen atoms segregated at the grain boundaries in coarse-grained Ti-0.3O microstructure.

Therefore, we try to consider another scenario for the physical background of the strong Fe-O attractive interaction at the grain boundary. We believe that the strong Fe-O attractive interaction can be attributed to a strong electronic binding due to specific local configurations at the grain boundary. That is to say, the configurational variation of substituted Fe and interstitial O is very limited within the specific bond-length and bond-angle in the "perfect crystal". On the other hand, a wide variety of configurations can be taken around the "grain boundary region" in view of the degrees of freedom in length and angle, which makes it possible to form more stable configurations. Accordingly, we concluded that it is reasonable to explain that could be very stable configurations of Fe-O only at the grain boundary.

In addition, we assume that there could be thermodynamically stable configuration of Fe-O pair at grain boundaries, where Fe has a variety of oxides and carbides with different crystal structure. This prediction could be reasonable in view of a recent published paper [*L. Ventelon, Phys. Rev. B 91 (2015) 220102(R)*], in which the Fe-C bonds around a screw dislocation was found to have anomalous stable structures due to cementite-like configurations. That finding indicates that the locally-distorted configuration sometimes creates significantly strong binding unlike that in the perfect crystal. While the local structure at grain boundaries is quite complicated and a universal discussion is still difficult, we believe that the grain boundary segregation of alloying elements which have stable oxides and carbides induces very stable substitutional-interstitial pairs unlike that in the perfect crystal region.

We feel that this would be an interesting topic which can be solved by machine learning technique using a wide variety of length and angle as feature values. We truly appreciate the reviewer's thought-provoking remarks on this topic. We also added one relevant sentence in the revised manuscript as follows:

(Page 9)

'Presumably, there are more stable configurations of Fe-O pairs at grain boundaries than in the bulk.'

(2) Knowing Fe-O interaction is so strong at the grain boundary, the authors still calculating only the separation energy of O in the absence of Fe. It does not make sense to me.

Response to reviewer's comment (2): We truly appreciate the reviewer for this nice comment. We totally agree with the reviewer's comment. Indeed, we have calculated a few cases of the cleavage energy when Fe-O pair is around the grain boundary region. At first, as we did not obtain statistically sufficient data points, we decided not to include this data. Now, we calculated the cleavage energy using additional configurations and added the results to Fig. 5h, where 500 eV was used for the energy cutoff according to comment (4). It is found that the change in cleavage energy in the case of Fe-O pair at GB becomes lower than in the case of O alone at GB, although in both cases the segregation (either in the form of Fe-O pair or O alone) lowers the GB cohesivity/binding, i.e. embrittles the grain boundary. We updated Fig. 5h and added relevant descriptions in the revised manuscript as follows:

(Page 9)

Furthermore, it should be noted that the change in the cleavage energy when considering Fe-O pair segregation at the grain boundary became lower than in the case of considering O segregation alone at the grain boundary, although in both cases the segregation tended to decrease the cleavage energy, i.e. deteriorate the grain boundary cohesivity.

(3) When the authors cleave the crystal along the grain boundary by inserting a vacuum, they leave all O atoms in one half and generate a clean Ti surface for the other. Maybe this is energetically reasonable, but should be justified carefully.

Response to reviewer's comment (3): We appreciate the reviewer for pointing this out and sorry for the confusion. In Fig. S11, we intended to show all candidate positions for interstitial atom around the grain boundary using the Voronoi vertex. Then, we performed the calculation using one O atom, instead of multiple O atoms at the grain boundary. We also agree that we need to search for energetically stable configurations when we consider the segregation of multiple O atoms case. To avoid confusion, the figure caption for Fig. S11 has been modified as follows:

(Page 33)

Fig. S11 DFT calculation set-ups of the grain boundary segregation and cleavage energies. All possible candidates of initial configuration for the interstitial oxygen sites at the grain boundary can be determined by Voronoi polyhedron analysis. DFT calculation was then carried out to explore the stable configuration for each interstitial site.

(4) The energy cut-off employed in the calculations was 450 eV. I would suggest 500 eV for O, for a high-quality result.

Response to reviewer's comment (4): We appreciate the reviewer for this comment. According to the suggestion, we re-calculated all the configurations using 500 eV for energy cutoff. The comparisons of the interaction energy, the segregation energy, and the change in cleavage energy using different energy cutoff are shown below (Fig. C). While most of the configurations provided approximately the same energies, a slight difference was found for Fe-O interaction depending on the energy cutoff. Accordingly, we replaced all images

in Fig. 5h of the revised manuscript using the new calculation data (500 eV cut-off energy) for a high-quality result. We appreciate the reviewer’s suggestion. The relevant description in the Experimental Method section of the revised manuscript has been modified as follows:

(Page 18)

‘A cut-off in plane-wave energy of 500 eV was applied using a first-order Methfessel–Paxton scheme that employed a smearing parameter of 0.2 eV.’

Fig. C The comparisons of calculation results (including interaction energy, segregation energy and change in cleavage energy) using different cut-off energies (blue: 450 eV, orange: 500 eV)

(5) The term “separation energy” is somewhat misleading. It was supposed to mean the effect of an impurity or alloying element on the cleavage energy along the grain boundary, but its literal meaning to many readers in this field should be the cleavage energy. So, what the authors really mean by this term is the change of separation energy.

Response to reviewer’s comment (5): We appreciate the reviewer for pointing out this misleading terminology in our manuscript. We totally agree with the reviewer that using ‘cleavage energy’ instead of ‘separation energy’ is more accurate, and thus can avoid confusing. Accordingly, we modified the related descriptions in the main text, figure caption of Fig. 5 and those in the Experimental Methods part, as follows:

(Page 8)

‘The effect of segregation on the cleavage behavior was subsequently calculated, where $\Delta\gamma_{int}$ is defined as the difference in the cleavage energies between Ti-O and pure Ti. The value of $\Delta\gamma_{int}$ is an excellent indicator to estimate the effect of oxygen on the ideal fracture toughness of the interface [22].’

(Page 19)

‘Accordingly, the difference in the ideal work of interfacial cleavage between alloys and pure metals...’

(6) In Fig. 4h, left panel, segregation energy of an impurity is presented in unit of eV/atom, but in the right panel the effect on cleavage energy is in unit J/m². It is better to use the same unit for readers’ convenience in case they want to compare these two quantities.

Response to reviewer’s comment (6): We appreciate the reviewer for this comment. As the reviewer suggested, the two energies (segregation energy and cleavage energy) can be linked using the same units, and there is an

advantage to using the same unit.

However, in the present calculations, the segregation energy is expressed as the segregation energy of individual atoms using eV/atom unit, which is not related to the per-area value. (We did not discuss the saturation of a number of atoms, where the per-area value becomes an important parameter.) On the other hand, we use J/m² unit for the cleavage energy to make it easier to understand in the discussion from the perspective of mechanical science. At present, we believe that this might be an intuitive and easy-to-understand notation. We appreciate for the reviewer's kind understanding on this issue.

REVIEWERS' COMMENTS

Reviewer #1 (Remarks to the Author):

The authors have adequately addressed the concerns raised in the initial review. Their attention to all of the issues is appreciated. The manuscript is now acceptable to this reviewer.

Reviewer #2 (Remarks to the Author):

The authors have provided a reasonably thorough response to the reviewer concerns, and the revisions have resulted in key improvements to the manuscript. In lines 200-202, the authors have chosen to retain reference to coincident site lattice theory despite the high Sigma value of this boundary. The manuscript could be improved by simply stating the misorientation angle about the misorientation axis. This concern should not prevent the manuscript as-is, but should be considered if another revision is requested.

Reviewer #3 (Remarks to the Author):

I am very glad to see that the authors have adequately addressed all the comments and suggestions I made (and probably those from other reviewers). I trust the manuscript is now appropriate for publication in NC in its revised form.

REVIEWERS' COMMENTS

Reviewer #1 (Remarks to the Author):

The authors have adequately addressed the concerns raised in the initial review. Their attention to all of the issues is appreciated. The manuscript is now acceptable to this reviewer.

Response to reviewer #1: We truly appreciate the interest that the referees have taken, as well as their valuable comments, which are greatly helpful to substantially improve the quality of our manuscript.

Reviewer #2 (Remarks to the Author):

The authors have provided a reasonably thorough response to the reviewer concerns, and the revisions have resulted in key improvements to the manuscript. In lines 200-202, the authors have chosen to retain reference to coincident site lattice theory despite the high Sigma value of this boundary. The manuscript could be improved by simply stating the misorientation angle about the misorientation axis. This concern should not prevent the manuscript as-is, but should be considered if another revision is requested.

Response to reviewer #2: We truly appreciate the interest that the referees have taken, as well as their valuable comments, which are greatly helpful to substantially improve the quality of our manuscript.

We have modified the relevant description of $(10\bar{1}4)$ boundary using the misorientation angle about the misorientation axis, as the reviewer kindly suggested.

(Page 8)

The $(10\bar{1}4)$ grain boundary with a misorientation angle of 50.5° about the $\langle 11\bar{2}0 \rangle$ rotation axis, which are energetically unstable compared to the other twin and grain boundaries from preliminary calculations...

Reviewer #3 (Remarks to the Author):

I am very glad to see that the authors have adequately addressed all the comments and suggestions I made (and probably those from other reviewers). I trust the manuscript is now appropriate for publication in NC in its revised form.

Response to reviewer #3: We truly appreciate the interest that the referees have taken, as well as their valuable comments, which are greatly helpful to substantially improve the quality of the theoretical calculations of our manuscript.